# Universal Semantic Disentangled Privacy-preserving Speech Representation Learning

## Abstract

The use of audio recordings of human speech to train LLMs poses privacy concerns due to these models' potential to generate outputs that closely resemble artifacts in the training data. In this study, we propose a speaker privacy-preserving representation learning method through the Universal Speech Codec (USC), a computationally efficient encoder-decoder model that disentangles speech into: (*i*) privacy-preserving semantically rich representations, capturing content and speech paralinguistics, and (*ii*) residual acoustic and speaker representations that enables high-fidelity reconstruction. Extensive evaluations presented show that USC's semantic representation preserves content, prosody, and sentiment, while removing potentially identifiable speaker attributes. Combining both representations, USC achieves state-of-the-art speech reconstruction. Additionally, we introduce an evaluation methodology for measuring privacy-preserving properties, aligning with perceptual tests. We compare USC against other codecs in the literature and demonstrate its effectiveness on privacy-preserving representation learning, illustrating the trade-offs of speaker anonymization, paralinguistics retention and content preservation in the learned semantic representations.

## 1 Introduction

The present and near-future of Generative AI (GenAI) revolve around foundational multimodal models (Achiam et al., 2023; Anil et al., 2023; Dubey et al., 2024). The extraordinary capabilities of Large Language Models (LLMs) as multimodal learning machines have ushered in a new paradigm for what GenAI can offer to our world (Barrault et al., 2023). These foundational LLMs are data-hungry, requiring massive amounts of multimodal training data. Speech and audio are essential modalities for many applications, and multimodal models require exposure to them during their training process (Borsos et al., 2023). Speech is a form of individual information (Nautsch et al., 2019), and the development of new foundational speech-aware models demands access to massive amounts of speech data to fully unlock their learning potential. The research community has collected and curated public data over the past decades, which has been used for specialized speech models (Łajszczak et al., 2024). However, in the realm of Responsible AI, every individual and organization must make proper use of individuals' data when training such foundational models, regardless of its public availability. Hence, it is imperative to develop privacy-preserving methods that enable advancing the state-of-the-art of foundational speech models trained on research speech data in a manner that safeguards individual privacy.

Foundational LLMs trained on language modeling tasks model the likelihood of generating coherent text sequences from a distribution of discrete tokens (Touvron et al., 2023). This allows them to produce expressive and varied responses during generation. Incorporating continuous signals, such as speech, into multimodal training objectives presents representation challenges that are circumvented by discretizing the distribution of the continuous space (Oord et al., 2017). Consequently, the model prediction quality is constrained by how well the target representations encode information from the data (Yu et al., 2024). For expressive and natural-sounding speech modeling, these representations

are required to capture rich semantic information, including content and paralinguistic information (such as prosody and sentiment) (Yang et al., 2024). However, from a speaker privacy perspective, they should not encapsulate any speaker-specific characteristics that could enable individual identification. We refer to these as semantic speaker privacy-preserving representations, which aim to capture the maximum semantic information while disentangling it from the speaker's identity. As illustrated in Figure 1, generating natural-sounding secure speech requires modeling these privacy-preserving semantic representations combined with a controlled reference speaker representation.

In this study, we present the Universal Speech Codec (USC), an encoder-decoder audio codec architecture that tokenizes speech into privacy-preserving discrete representations tailored for speech-aware LLMs. USC simultaneously learns semantically meaningful discrete representations that capture the speech content and paralinguistics such as pacing, emphasis, and sentimental aspects, while also learning the additionally required speaker residual representations, necessary for reconstructing the original waveform. Motivated by the work of Zhang et al. (2024), we introduce a speaker privacy-preserving representation learning method with enhanced paralinguistic and anonymization biases. In addition to incorporating a semantic distillation, we include a specific speaker classifier gradient reversal (Martín-Cortinas et al., 2024), the usage of Local Differentiable Privacy (LDP) (Shamsabadi et al., 2022), and the quantizer dropout technique (Kumar et al., 2023) to further bias the semantic representations. To the best of our knowledge, USC is the lowest bit-rate high-fidelity speech codec in the literature for larger context windows and scalable secure speech-aware LLMs.

We benchmark our technique against four open-source alternatives through objective evaluations and show that the proposed USC's semantic representations have good content preservation and low speaker-specific characteristics while encoding a huge amount of paralinguistic sentiment speech information. Moreover, the residual representations augments the semantic ones with the remaining speaker attributes, reporting state-of-the-art metrics on speech waveform reconstruction. Additionally, we define a new set of metrics and requirements to assess the privacy-preservation of the learned speech semantic representations through the $k$-anonymity factor for speech. This test is based on the concept that an individual achieves $k$-anonymity if their reconstructed speech is indistinguishable from at least $k$-1 other individuals within the dataset. To corroborate the presented test, we further conduct human perceptual evaluations to validate the correlation between the proposed objective privacy-preserving test and human perception. We summarize our contributions as:

1. We propose a speaker privacy-preserving representation learning method based on the USC architecture, that disentangles speech semantics from speaker-identifiable traits, surpassing all available baselines in jointly encoding content and paralinguistic information.

2. We present an ensemble of speaker disentanglement techniques and demonstrate that Local Differential Privacy can be scaled for speaker privacy-preserving representation learning.

3. We introduce a privacy-preserving evaluation that defines a set of metrics to assess the level of anonymization in speech representations, which is validated by human perceptual tests.

Finally, we show USC's effectiveness by presenting an LLM-based Text-To-Speech (TTS) model trained on USC tokens. We validate the presented representation learning methodology enabling Voice Conversion (VC) through a novel semantic Partial-Teacher-Forcing (PTF) technique in Appendix A. Without being trained specifically for this task, the presented model can generate the target speaker's voice while preserving the paralinguistic characteristics of the source speaker.

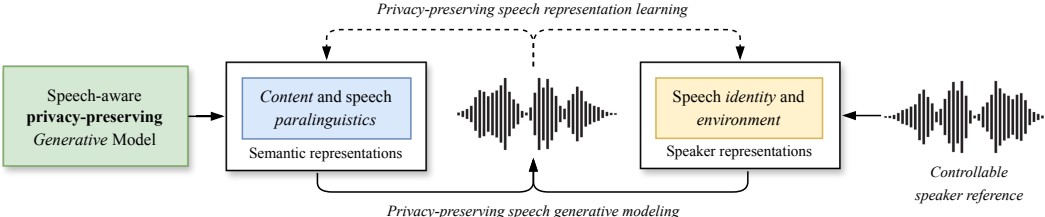

Figure 1: General scheme for privacy-preserving speech representation learning and generative modeling. Dashed lines denote the extraction of privacy-preserving speech features. Solid lines showcase the process of secure generative speech by modeling semantincs and controlling the identity.

## 2 RELATED WORK

**High fidelity audio discretization**: neural discretization was pioneered through the Vector Quantized-Variational Autoencoder (VQ-VAE) by Oord et al. (2017). One of the early adaptations of VQ-VAE for audio discretization was introduced by Gârbacea et al. (2019), where they replaced the VQ-VAE convolutional decoder in the original architecture with a decoder based on the auto-regressive WaveNet (Oord et al., 2016) vocoder. This work introduced a vocoder reconstruction loss into the learning of discrete audio representations. Parallel to the developments in audio discretization, neural vocoding techniques evolved into fully parallel convolutional high-fidelity GAN-based vocoders. HiFi-GAN (Kong et al., 2020) was the first parallel GAN-based vocoder to outperform WaveNet in both efficiency and quality, establishing GAN-based architectures as the preferred approach for neural vocoding. SoundStream (Zeghidour et al., 2021) presented an audio codec based on a convolutional encoder and decoder trained in a GAN setup. It introduced the VQ-GAN (Esser et al., 2021) formulation into the audio discretization domain and proposed the Residual Vector Quantizer (RVQ), which significantly increased the discretization capabilities and generalization towards universal audio codec models. EnCodec (Défossez et al., 2023) improved the SoundStream recipe through a loss balancer approach and introducing a HiFi-GAN based decoder. In the vocoding domain, BigVGAN (Lee et al., 2023) improved the HiFi-GAN recipe adding a periodic inductive bias using the Snake activation (Ziyin et al., 2020) and improved discriminators. Building upon the EnCodec and BigVGAN, Descript Audio Codec (DAC) (Kumar et al., 2023), scaled to support 44.1kHz and presented an improved RVQ learning process through quantizer dropout.

**Disentangled Speech Representations**: Contrary to general audio, speech can be subdivided into different attributes (Polyak et al., 2021): Content represents the main information in the speech. The speaker identity corresponds to the specific characteristics of the speaker. Paralinguistic information encompasses prosodic elements such as intonation, stress or pace. Acoustic details refer to any extra environmental information present in the speech signal. Large-scale speech disentanglement models are based on large pre-trained Self-Supervised Learning (SSL) models (Hsu et al., 2021) which disentangle speech attributes through different hidden layers (Yang et al., 2021), used for supervised downstream tasks like Speech Recognition and Speaker Identification (Chen et al., 2022).

**Speech Codecs**: Speech-specific codecs leverage the disentanglement capabilities of speech. Disen-TF-Codec (Jiang et al., 2023) presents a way to extract a temporal pooled timbre representation to disentangle speaker from content. FACodec (Ju et al., 2024) proposes a factorized codec to perform speech attribute disentanglement through information bottleneck, speaker gradient reversal, and supervised training signals. On the other hand, other solutions use SSL models to learn a disentangled tokenization. RepCodec (Huang et al., 2024) directly learns a tokenization layer on top of a specific selection of SSL model layers. USM AudioPalm (Rubenstein et al., 2023) tokenizes a SSL model through training a tokenizer on Speech Recognition downstream tasks to learn content-rich representations. SpeechTokenizer (Zhang et al., 2024) distills semantic information from a pre-trained SSL model to bias the first codebook of the RVQ to encode the speech content without requiring transcripts. NPU-NTU (Yao et al., 2024) proposes, in addition to semantic, F0 distillation to further RVQ tokens to hierarchically capture content, sentiment and speaker in separate representations.

## 3 METHOD

The proposed model, illustrated in Figure 2, is based on a modified version of the DAC model (Kumar et al., 2023). It encodes speech into discrete residual representations and decodes them back to the reconstructed waveform. The disentanglement modules bias the representations to encode semantically rich features without speaker-specific traits. Only the first codebook is biased to obtain a single set of non-residual semantic tokens. Exact model details are depicted in Appendix B.

### 3.1 UNIVERSAL SPEECH CODEC

**Encoder & Decoder**: The encoder performs temporal downsampling of the input waveform $x$ through a series of strided and residual convolutional blocks to obtain the encoded representations $z_e$ of the input. The decoder mirrors the encoder architecture and reconstructs the waveform $\hat{x}$ from the quantized representation $z_q$ of $z_e$. In both modules, we replaced the traditional Snake activation function with the log-scale Snake-beta (Ziyin et al., 2020). In the decoder, we removed the final *tanh* activation, as it introduced harmonic distortions into the generated speech (Evans et al., 2024).

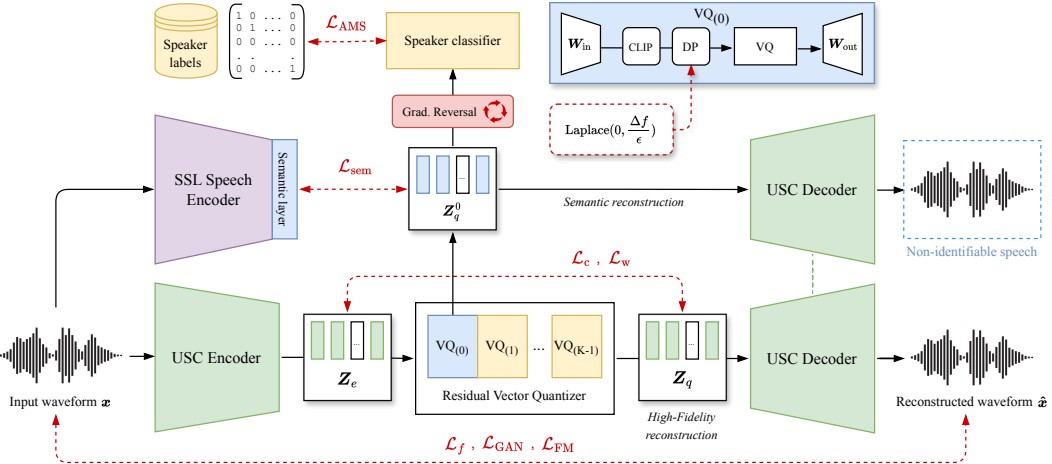

Figure 2: USC architecture. Red dashed lines denote training objectives while black continuous lines refer to the inference pipeline for high-fidelity and semantic reconstruction of speech.

**Residual Vector Quantizer**: In RVQ, multiple $K$ Vector Quantizers (VQ) are employed in a hierarchical manner to achieve a more fine-grained quantized representation $z_q$ of the input latent $z_e$ (Zeghidour et al., 2021). The input vector is first quantized using the initial quantizer $\mathrm{VQ}_{(0)}$, and the difference between the input and the quantized representation is then recursively discretized using the subsequent codebooks. RVQ provides the flexibility to choose the number of codebooks to use, thus for a variable quantizer number $n \leq K - 1$, the discretized representation can be obtained by:

$$z_q^n = z_q^0 + \sum_{i=1}^{n} \mathrm{VQ}_{(i)}(z_e - z_q^{i-1}) \tag{1}$$

and $z_q^0$ is the first non-residual quantized representation of $z_e$, i.e. $z_q^0 = \mathrm{VQ}_{(0)}(z_e)$. The derivation of Equation 1 is presented in Appendix C. For simplicity, we denote the final quantized latent $z_q^{K-1}$, which uses all $K$ available quantizers in the RVQ, as $z_q$.

For the RVQ component in USC, we employ factorized and $L2$-normalized codes introduced by Yu et al. (2022), which include an input $W_{\mathrm{in}}$ and output $W_{\mathrm{out}}$ projection before and after the quantization step, to improve the codebook usage across all residual quantizers.

## 3.2 Speaker reversal

The speaker reversal module is the first component responsible for removing speaker-specific information from the speech semantic representations. It consists of a cross-entropy based speaker classifier and a gradient reversal layer. The speaker classifier is trained to identify the speaker from the semantic representations. Then, we reverse the computed gradients during backpropagation (Ganin & Lempitsky, 2015) to suppress relevant information used for the speaker identification task.

The gradient reversal component and speaker classifier are based on the work by Martín-Cortinas et al. (2024), which uses a stack of transformer encoder layers as a speaker extractor. However, instead of training on a contrastive objective, the speaker classifier's output is projected to a finite number $N$ of known speakers, and it is trained with the AMSoftmax loss, $\mathcal{L}_{\mathrm{AMS}}$ (Wang et al., 2018), using mean reduction over the $n$ batch elements:

$$\mathcal{L}_{\mathrm{AMS}} = -\frac{1}{n} \sum_{i=1}^{n} \log \left( \frac{e^{s(\boldsymbol{W}_i \cdot \boldsymbol{F}_i - m)}}{e^{s(\boldsymbol{W}_i \cdot \boldsymbol{F}_i - m)} + \sum_{j=1, j \neq i}^{N} e^{s(\boldsymbol{W}_j \cdot \boldsymbol{F}_i)}} \right) \tag{2}$$

where $\boldsymbol{W}$ is a learnable normalized weight, $\boldsymbol{F}_i$ is the normalized speaker $i$ output logit, $m = 0.4$ is the additive margin value and $s = 30$ is the constant scaling factor of the AMSoftmax loss function.

### 3.3 SEMANTIC DISTILLATION

Only biasing the semantic representations with the speaker reversal component leads to heavy degradation of meaningful semantic information. The easier solution for the model to converge is to destroy as much information from the semantic codebook to remove speaker-specific information. Therefore, following Zhang et al. (2024), we introduce a bias in the representations via semantic distillation of a pre-trained Self-Supervised Learning (SSL) speech model (Mohamed et al., 2022).

As a semantic teacher, we choose the multilingual version of HuBERT (Hsu et al., 2021) to extract the semantic targets $\boldsymbol{S} = \{\boldsymbol{s}^{(0)}, \cdots, \boldsymbol{s}^{(L)}\}$ for the $L$ transformer blocks. We apply a modified version of the continuous DistillHuBERT loss (Chang et al., 2022) as the distillation function directly on the semantic representations $\boldsymbol{z}_q^0$. Instead of using the original log-sigmoid activation of cosine similarity, we use the Cosine Embedding Loss. The semantic distillation loss $\mathcal{L}_{\text{sem}}$ follows:

$$\mathcal{L}_{\text{sem}} = \lambda_{L1}\mathcal{L}_{L1} + \lambda_{\cos}\mathcal{L}_{\cos} = \lambda_{L1}||\boldsymbol{z}_q^0 - \boldsymbol{s}^{(l)}||_1 + \lambda_{\cos} \max\left(0, 1 - \frac{\boldsymbol{z}_q^0 \cdot \boldsymbol{s}^{(l)}}{\|\boldsymbol{z}_q^0\|\|\boldsymbol{s}^{(l)}\|}\right) \quad (3)$$

where $\lambda_{L1}$ and $\lambda_{cos}$ are set to $0.15$ and $1$ respectively. We choose the layer $l = 9$ of HuBERT as it is shown to contain rich semantic information without speaker-identifiable traits (Chen et al., 2022).

### 3.4 QUANTIZER DROPOUT

We want to have paralinguistic information related to prosody and sentiment encoded into the learned semantic representations. The waveform reconstruction and perceptual loss, i.e. the vocoder loss, contains rich semantic details, but it is highly entangled with speaker-specific characteristics. To leverage this for learning semantic representations, we use quantizer dropout (Zeghidour et al., 2021), which enables variable bit-rate capabilities during training. Following the modified approach from Kumar et al. (2023), we apply quantizer dropout with a probability of $p = 0.5$. By doing so, we ensure that the decoder is able to reconstruct the waveform at different levels of the RVQ, and lead it to learn from the most to the least significant information with each additional residual quantizer.

However, the waveform reconstruction and perceptual losses are very speaker-based strong signals, so we limit its influence by stopping the gradients from propagating to the encoder when the dropout probability chooses to use only the semantic quantizer. By doing so, we solely train the decoder to reconstruct faithful speech from the semantic representations while propagating the perceptual loss to the decoder and the quantizer. The proposed method guarantees that the decoder is trained to faithfully reconstruct speech from the semantic representations, and encourages the semantic representations to capture relevant paralinguistic information for faithful speech synthesis.

### 3.5 LOCAL DIFFERENTIAL SPEAKER PRIVACY

The speaker gradient reversal technique does not guarantee that information is being reliably removed from the semantic representations for an unseen identity, as the classifier is trained on a limited set of labeled speakers. To ensure stronger guarantees of speaker information removal, we employ tools from Local Differential Privacy (LDP) (Shamsabadi et al., 2022) in the USC tokenization. LDP protects the privacy of individual records and provides strong theoretical guarantees on anonymization. We employ a widely used variant of Local Differential Privacy (LDP) known as the Laplace mechanism, which anonymizes a function $f$ by adding Laplace noise. The noise is controlled by a hyperparameter $\epsilon$ and the $L_1$ sensitivity of the function, $\Delta f$. We apply the Laplace noise block to the semantic quantizer (i.e. $\text{VQ}_{(0)}$) after the input projection of the factorized quantization block (see Figure 2). We clip the norm of the projection output to $C$, which results in an $L1$ sensitivity upper bounded by $2C$. The clipping value $C$ was estimated by computing the average of the $L_1$-norm over several training batches from a USC model without the Laplace noise block. During training we add the noise sampled as $n \sim \text{Laplace}(0, 2C/\epsilon)$ to the output of the down projection layer. The smaller the value of $\epsilon$, the more spread out the Laplace distribution is. The choice of hyper-parameter $\epsilon$ dictates the degree of privacy-utility tradeoff. Privacy can be quantified by speaker re-identification accuracy and utility is defined as speaker fidelity of the generated speech. During inference we simply omit the noise block (Chouchane et al., 2023). We provide extensive results on the impact of using LDP in Section 4.3 for the privacy-preserving evaluation results.

### 3.6 Training objectives

**Reconstruction Loss**: The reconstruction loss $\mathcal{L}_f$ follows the approach proposed in Kumar et al. (2023). We employed a combination of multi-scale spectrogram losses to capture both coarse and fine-grained spectral characteristics. It is defined as the L1 distance between the multiple scales of mel-spectrograms from the predicted and target waveforms. Specifically, for $i \in \{5, 6, \ldots, 11\}$, we computed different mel-spectrograms in the decimal logarithmic scale using window sizes of $2^i$, with their corresponding hop lengths set to $2^i/4$, and the number of mel bins set to $5 \times i$.

**Perceptual loss**: We introduced a perceptual GAN-based loss proposed by Kumar et al. (2023). This is a combination of a Multi-Period Discriminator (MPD) and a Multi-Band Multi-Resolution STFT Discriminator (MB-MRSD). The MPD operates on the waveform signal, where each discriminator reshapes the waveform into a two-dimensional representation with varied heights and widths to capture multiple periodic structures. The MB-MRSD operates in the frequency domain. Each subset of multi-resolution discriminators converts the waveform into different complex STFT resolutions. Then, each STFT is split into different subbands to train a specific resolution discriminator per band. This approach alleviates the aliasing of high frequencies. We use the least squares adversarial loss (Mao et al., 2017), $\mathcal{L}_{\text{GAN}}$, and the L1 feature matching loss, $\mathcal{L}_{\text{FM}}$, which minimizes the distance for every intermediate feature of the discriminator layers between real and generated waveform.

**Codebook learning**: The RVQ is trained with both commitement $\mathcal{L}_w$ and codebook usage $\mathcal{L}_c$ loss functions with straight-through gradient estimation (Oord et al., 2017). The commitment loss encourages the encoder's output to be close to the quantized value in the codebook. The codebook loss, on the other hand, encourages the codewords themselves to be updated and better represent the data distribution by minimizing the distance between the encoder's output and the assigned codeword.

Overall, USC is trained to optimize the next total $\lambda$-weighted balanced loss over a training batch:

$$\mathcal{L} = \underbrace{\lambda_f \mathcal{L}_f + \lambda_{\text{GAN}} \mathcal{L}_{\text{GAN}} + \lambda_{\text{FM}} \mathcal{L}_{\text{FM}}}_{\text{Reconstruction + Perceptual}} + \underbrace{\lambda_c \mathcal{L}_c + \lambda_w \mathcal{L}_w}_{\text{Codebook + Commitement}} + \underbrace{\lambda_{\text{AMS}} \mathcal{L}_{\text{AMS}} + \lambda_{\text{sem}} \mathcal{L}_{\text{sem}}}_{\text{Speaker disentanglement}} \quad (4)$$

## 4 Experiments and results

### 4.1 Experimental setup

**Datasets**: We used the same custom speech dataset as in Łajszczak et al. (2024), which consisted of more than 100K hours of public domain speech data in more than 5 different languages, with English being the dominant one. We added to this dataset a split of more than 1K different internal labeled studio-quality speakers. We ensured that 20% of the samples in a training batch were from this labeled set to train the speaker classifier. Speakers without labels did not apply a loss on the speaker classification. For objective evaluation, we used 10 different internal speakers with various expressive styles, including excited, cheerful, mindful, conversational, and long-form reading. For the privacy-preservation evaluation, we took a labeled pool of 7974 speakers from different sources.

**Training**: comprises two steps, first, a 16 kHz USC variant is trained from scratch for 1M steps, leveraging the maximum available speech data (Appendix D). Following Equation 4, the reconstruction, perceptual, commitment and codebook terms are weighted with $\lambda_f = 15$, $\lambda_{\text{GAN}} = 1$, $\lambda_{\text{FM}} = 2.0$, $\lambda_c = 1$ and $\lambda_w = 0.25$ following the unmodified weights of (Kumar et al., 2023). For the speaker biases we set $\lambda_{\text{AMS}} = 25$ as in (Martín-Cortinas et al., 2024) and $\lambda_{\text{sem}} = 45$ which is half the weight of Zhang et al. (2024). For the LDP, we set $\epsilon = 15$ which provided the best subjective privacy-utility trade-off. To produce high-quality speech, a 24 kHz decoder is trained with frozen encoder and RVQ on 24 kHz filtered data for 2.5M steps with reconstruction and perceptual losses. Both trainings use 3-second speech chunks. More training parameters are provided in Appendix E.

**Model**: USC encodes waveforms at 16 kHz with a temporal downsampling of $640\times$. Each encoded latent corresponds to 40ms of speech (a frame-rate of 25Hz). The frame-rate of USC is exactly half of the temporal dimension of the teacher semantic distiller model, thus we apply average pooling across the time dimension to get the semantic targets. We use a 6-layer RVQ to get the discretization $C_{0:5}$. We use 16,384 tokens in $C_0$ to encode a larger number of semantic variations. For the residual layers, we use 1024 tokens each. With all of that, USC achieves a bit-rate of 1.6 kbps for all the discretized tokens and a bit-rate of 0.35 kbps for the semantic representations (Appendix F).

## 4.2 EVALUATION METRICS

We evaluate USC against four neural codecs: EnCodec (Défossez et al., 2023), DAC (Kumar et al., 2023), SpeechTokenizer (Zhang et al., 2024), and FaCodec (Ju et al., 2024). Our metrics are inspired by the VoicePrivacy Challenge (VPC) (Tomashenko et al., 2024) privacy and utility evaluation.

For privacy metrics, we measure the retention of speaker-identifiable traits (SIM) through a state-of-the-art speaker verification model. We extract speaker embeddings from the pre-trained TitaNet model (Koluguri et al., 2022) and compute the cosine similarity score to provide an objective distance measurement of speech identity (Dehak et al., 2010).

For utility metrics, we measure content and sentiment preservation. For content, we evaluate the Word Error Rate (WER) by transcribing the resynthesized speech using the Whisper v2-large model (Radford et al., 2023) and the Short-Time Objective Intelligibility (STOI), which evaluates the intelligibility of the signal in the presence of noise or other distortions (Taal et al., 2010). For sentimental information, we evaluate the Concordance Correlation Coefficient (CCC) through a proprietary sentiment extractor based on Wav2Vec2-XLSR (Baevski et al., 2020), fine-tuned on an internal dataset of 180 hours of multi-speaker, labeled spontaneous speech. We provide the correlation metric between the outputs of the sentiment logits to quantify its preservation (Atmaja & Akagi, 2021). Additionally, to measure the intonation faithfulness, we provide the F0 Spearman's Correlation Coefficient (SCC) (Spearman, 1961) to measure the monotonic non-absolute pitch correlation.

To report quality metrics of the reconstructed speech, we report the ViSQOL v3 Speech (Chinen et al., 2020) and the Perceptual Evaluation of Speech Quality (PESQ) (Rix et al., 2001) metric.

## 4.3 PRIVACY-PRESERVING TEST: LINKABILITY AND SINGLING OUT

Completely eliminating speaker-specific traits while retaining paralinguistic richness is a conflicting task (Cai et al., 2024). Certain paralinguistic aspects are characteristic traits that facilitate speaker identification, yet they are crucial to be preserved for natural-sounding and expressive speech modeling. Motivated by this, we assess the level of privacy-preservation in our speech semantic representations through the introduction of a speech privacy-preserving test based on the $k$-anonymity metric (Samarati & Sweeney, 1998). $k$-anonymity is a property of data that guarantees that the information for each person contained in a set cannot be distinguished from at least $k-1$ other individuals in the same set. This allows the preservation of certain aspects of the voice, without revealing the individual's identity. We define two metrics based on $k$-anonymity to assess linkability and singling out (Cohen & Nissim, 2020), adhering to the European Union anonymization techniques ($EU$, 2014).

**Linkability**: ability to link two anonymized speech samples pertaining to the same individual.

**Singling out**: ability to locate an individual's sample within the dataset. Even if the anonymized speech retains some original characteristics, it should not be possible to isolate the original speaker.

Consider a dataset $\mathcal{D}$ with speech utterances from a set $\mathbb{S} = \{s_1, \ldots, s_N\}$ of $N$ speakers. The dataset is split into two partitions, the reference dataset $\mathcal{D}_r$, and the evaluation dataset, $\mathcal{D}_e$, each of them including recordings from all the $N$ speakers. For each speaker $s \in \mathbb{S}$, we run $L$ speaker identification tests, comparing a random speaker utterance $x_s^l \in \mathcal{D}_e$ with $N$ utterances, one for each speaker, $Y_s^l = y_{s_1}^l, \ldots, y_{s_N}^l \in \mathcal{D}_r$, randomly selected for the $l$ test.

We calculate the speaker similarity between the evaluation utterance $x_s^l$ and the $N$ reference utterances in $Y_s^l$ across the $L$ test cases. The similarity measurement relies on the SIM metric presented in Section 4.2. We use automatic metrics, as they have proven more accurate than humans for speaker identification (Kahn et al., 2011). Then, we compute the classification rank, $r_s^l$, defined as the position in the descending list of similarities of the utterance from the same speaker. Finally, we compute the mean rank per speaker, $\bar{r}_s$ as the average of $r_s^l$ across the $L$ tests:

$$\boldsymbol{r}_s^l = \text{rank}|_{N_\downarrow}(\text{sim}(x_s^l, y_{s_n}^l)) \in \mathbb{N}^{L \times N}, \quad \bar{\boldsymbol{r}}_s = \frac{1}{L} \sum_{l=1}^{L} \boldsymbol{r}_s^l \in \mathbb{R}^N \quad (5)$$

Having an average rank $\bar{r}_s \geq k$ for speaker $s$ means that, on average, there are audio samples from at least $k-1$ different speakers which are more similar than other samples from the same speaker. For non-anonymized speech (the dataset $\mathcal{D}$ contains original speech recordings) and a perfect similarity

metric, the rank would be 1. For completely indistinguishable samples, random guessing would generate ranks that follow a uniform distribution over the possible N rank options. Therefore, the expected value of the uniformly distributed classification rank would be $\mathbb{E}[\boldsymbol{r}_s] = (N+1)/2$. Further distribution analysis and percentile computations for random guessing are provided in Appendix H.

To report linkability (ability to link anonymized samples), the similarities are computed using the anonymized version of the datasets $\mathcal{D}_r$ and $\mathcal{D}_e$ with anonymized utterances. For the singling out metric (ability to locate individuals in anonymized dataset), the similarities compare the anonymized version of the reference dataset $\mathcal{D}_r$ and the version of $\mathcal{D}_e$ with the original recordings.

**Perceptual privacy evaluations**: We introduce this extra evaluation involving human preference to check if the proposed test, based on objective measurements, is correlated with human perception. We have selected to validate the singling-out scenario as it poses the greatest challenge for privacy preservation, where the ability to pinpoint the original speaker is the most critical privacy risk.

We have randomly selected 20 unique speakers and built 20 $A/B/X$ triplets selected as:

$X$: Unidentifiable speech sample (semantic reconstruction, $C_0$ of the USC)
$A$: Speech sample (utterance with different content of same speaker)
$B$: Speech sample (utterance with different content from a a similar speaker).

The listeners are asked to identify which speaker ($A$ or $B$) is the one that generated the semantic reconstruction $X$. To get the $B$ samples from similar speakers, we first identify a pool of speakers who got a higher objective singling out ranking than the original speaker. Then, we randomly select $B$ samples from each speaker's pool. A test case example is shown in Appendix J.

## 4.4 Results

Table 1: Evaluation metrics on a dataset of 1200 samples for 10 different internal speakers with varied expresive speaking styles. The best score is highlighted in bold. If there is no statistically significant difference between best scores ($p_{\text{value}} > 0.05$), multiple systems are highlighted.

| Model | RVQ | BW | WER ↓ | STOI ↑ | PESQ ↑ | ViSQOL ↑ | SIM ↑∥↓ | CCC ↑ | SCC ↑ |
|---|---|---|---|---|---|---|---|---|---|
| Recordings | - | - | 0.053 | 1.000 | 4.500 | 5.000 | 1.000 | 1.000 | 1.000 |
| *High Fidelity Reconstruction* | | | | | | | | | |
| EnCodec | $C_{0:7}$ | 6.00 kbps | **0.056** | 0.943 | 2.327 | 3.686 | 0.802 | 0.914 | 0.891 |
| DAC | $C_{0:8}$ | 7.75 kbps | 0.059 | **0.975** | **3.311** | **3.975** | **0.910** | **0.969** | **0.962** |
| SpeechTokenizer | $C_{0:7}$ | 4.00 kbps | **0.057** | 0.925 | 2.332 | 3.539 | 0.811 | 0.915 | **0.957** |
| FaCodec | $C_{0:5}$ | 4.80 kbps | **0.056** | 0.956 | 2.724 | 3.566 | 0.864 | 0.951 | 0.961 |
| USC | $C_{0:5}$ | 1.60 kbps | **0.056** | 0.958 | 2.991 | 3.706 | 0.884 | **0.957** | 0.959 |
| *Semantic Reconstruction* | | | | | | | | | |
| EnCodec | $C_0$ | 0.75 kbps | 0.226 | 0.776 | 1.147 | 1.786 | 0.145 | 0.433 | 0.641 |
| DAC | $C_0$ | 0.86 kbps | 0.171 | **0.785** | **1.195** | **2.077** | 0.248 | 0.440 | 0.728 |
| SpeechTokenizer | $C_0$ | 0.50 kbps | 0.077 | 0.630 | 1.101 | 1.095 | **0.056** | 0.273 | 0.118 |
| FaCodec | $C_{0:2}$ | 2.40 kbps | **0.067** | 0.714 | 1.086 | 1.632 | 0.313 | **0.629** | **0.815** |
| USC | $C_0$ | 0.35 kbps | 0.091 | 0.685 | 1.067 | 1.687 | 0.218 | 0.526 | 0.526 |

**Objective evaluation**: Table 1 summarizes the evaluation results. USC achieves competitive performance in *High-fidelity reconstruction* in both PESQ and ViSQOL, outperforming SpeechTokenizer and FaCodec while reducing its bit-rate by 60% and 80% respectively for 24 kHz waveform reconstruction. DAC slightly outperforms all baselines for high-fidelity reconstruction, potentially due to its balanced 44.1 kHz data selection. Regarding *Semantic reconstruction*, SpeechTokenizer achieved the best speaker similarity metric, which demonstrates better anonymization characteristics. Figure 3 reveals that SpeechTokenizer reconstructs completely inexpressive speech, destroying all paralinguistic information. FaCodec, while conditioned on a mean average speaker embedding (Yao et al., 2024), does not modifiy drastically the semantic waveform compared to the original recording, thus showcasing some speaker leakage in its independent content $VQ_{(0)}$ and prosody $RVQ_{(1:2)}$, $C_{0:2}$. USC, on the other hand, recovers a structured waveform with shifted pitch harmonics, generating different identities across the same utterance. These identity shifts within a word/sentence are out-of-distribution samples for speech intelligibility systems, resulting in USC reporting a higher

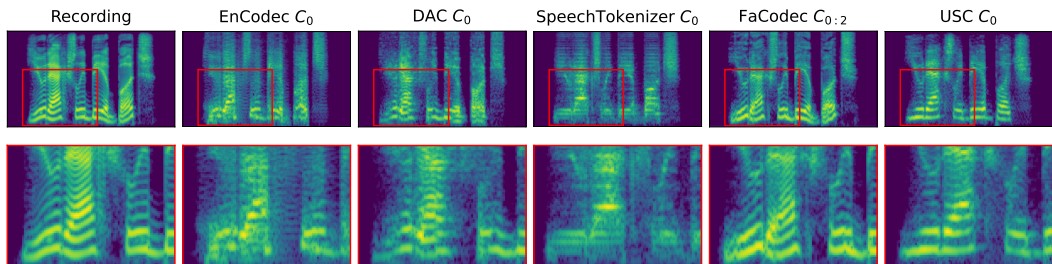

| Recording | EnCodec $C_0$ | DAC $C_0$ | SpeechTokenizer $C_0$ | FaCodec $C_{0:2}$ | USC $C_0$ |

Figure 3: Spectrogram visualization of semantic $C_0$ reconstruction of a speech sample from all the compared baselines, with a zoomed in view of the pitch harmonics. More examples in Appendix I.

WER and lower STOI metric, thus falling behind SpeechTokenizer and FaCodec in content preservation. Preserving paralinguistic features effectively leads to increased speaker similarity, as certain prosodic characteristics facilitate speaker identification. This observation is further corroborated by the CCC sentiment and the F0 SCC metric, where USC closes the gap from SpeechTokenizer's semantic representations by 47.97% and 46.25% respectively, but falls behind FaCodec, whose semantic reconstructions are the closest to the recordings at the cost of a $6.85\times$ larger bit-rate. A further analysis of pitch contour is shown in Appendix G. EnCodec and DAC do not apply any disentanglement, their $C_0$ reconstructions are low-quality acoustic versions of speech, reporting high F0 SCC metric but high WER (Figure 3).

**Privacy evaluations**: Following the proposed privacy-preserving speech test, we prepared a dataset of $N = 7974$ speakers. This dataset is split into the reference ($\mathcal{D}_r$) and evaluation ($\mathcal{D}_e$) sets, each of them with 45 utterances per speaker. The average duration of each sample is 6 seconds. To compute the mean rank per speaker $\bar{r}_s$, we used $L = 100$ tests. For this evaluation, we validated two variants of USC: with and without LDP applied on the learned semantic representations to assess the impact of using LDP in the speaker privacy-preserving task. Table 2 shows the ranking distributions of the linkability and singling out across all the evaluated speakers. We report the median (p50) and the first percentile (p1). We define the latter as the speech $k$-anonymity factor, and it illustrates the worst-case for privacy preservation, as it represents the minimum number of speakers who are indistinguishable from the speakers with the most unique representation within the evaluation dataset.

For *Linkability*, when USC is not trained with LDP, 50% of the anonymized speakers (p50) are not distinguishable from at least 495 other anonymized speakers. For the final USC variant with LDP, this number scales to 1029. Focusing on the first percentile (p1), which we name as the speech $k$-anonymity factor, we show that for 99% of the speakers, there are at least 35 indistinguishable anonymized speakers for the variant without LDP and 159 for the final USC variant. This result shows that adding LDP improved the linkability metric by 368% relatively to not using LDP. Regarding *Singling Out*, when using the final USC with LDP, 50% of the anonymized speakers are not distinguishable from at least 816 other speakers in the dataset, while for 99% of the anonymized speakers, the $k$-anonymity factor, is 68 speakers that are closer than the original speaker identity. Again, this is a relative improvement of 508% compared to not using LDP.

Table 2: Percentiles p50 (median) and p1 ($k$-anonymity factor) for Linkability and Singling out.

| Model | *Linkability* | | *Singling out* | |
| --- | --- | --- | --- | --- |
| | **Rank p50** (median) | **Rank p1** ($k$-anonymity) | **Rank p50** (median) | **Rank p1** ($k$-anonymity) |
| Recordings | 1.01 | 1.00 | 1.01 | 1.00 |
| EnCodec | 435.61 | 37.20 | 673.63 | 32.10 |
| DAC | 266.66 | 12.04 | 181.74 | 4.52 |
| SpeechTokenizer | **1929.61** | **774.57** | **2459.81** | **601.68** |
| FaCodec | 465.12 | 41.72 | 414.54 | 14.15 |
| USC (w/o LDP) | 495.21 | 34.98 | 320.75 | 12.22 |
| USC | 1029.03 | 159.96 | 816.49 | 68.91 |
| Random (Theoretical) | 3987.50 | 3452.06 | 3987.50 | 3452.06 |

Compared to other baselines, the results align with the objective metrics. EnCodec and DAC do not apply speaker disentanglement, thus they report lower privacy-preserving linkability and singling out metrics. SpeechTokenizer reports the highest privacy-preserving metric of all, at the cost of destroying all paralinguistic information in its semantic representations. FaCodec reports quite low singling out metrics. As it is conditioned on a mean speaker embedding, this result suggests that, in a large-scale evaluation, FaCodec's content and prosody representations leak some of our evaluation speakers' information and thus are worse at preserving privacy for all the speakers in our pool.

We corroborate the objective results through the presented *perceptual privacy evaluation* test. The $A/B/X$ samples were evaluated by human raters using the click-worker crowd-sourcing platform. We evaluated the final version of USC with LDP. The analysis of the test shows that the probability of finding out that $X$ is the same speaker as $A$ is $0.51 \pm 0.02$ (Wilson confidence interval, at a 5% significance level). As expected, the test does not allow concluding that the speaker can be singled out from the anonymized speech. As a reference, we repeated the test, using the same samples but with original waveforms. In this case, the probability of detecting the speaker is $0.61 \pm 0.02$, showing that according to human testers, it is possible to identify the source speaker when non-anonymized speech is used. Note that a probability of $0.61$ may not seem high, but in addition of the noisy nature of crowd-sourcing data, $B$ samples are chosen from the most similar speakers, thus making the task non-trivial for a human listener.

## 5 CONCLUSIONS AND FUTURE WORK

In this research, we presented a method for speech disentanglement and speaker privacy-preserving representation learning. The method relies on the Universal Speech Codec (USC), a low-bit-rate speech codec that disentangles speech into two representation sets through its Residual Vector Quantization (RVQ) component. Firstly, the main codebook, $C_0$, learns rich semantic speech representations that encode speech content and paralinguistic information while preserving non-prosodical speaker privacy. Secondly, USC learns the complementary speaker-specific information to enable high-fidelity speech reconstruction in the residual codebooks. Through extensive evaluations, we showed that USC's semantic privacy-preserving representations encode a high level of content and sentimental information while being more efficient than any other baselines. Combining both representations, we show USC's state-of-the-art performance in achieving high-quality speech reconstruction. Additionally, we proposed a new speech privacy assessment protocol based on $k$-anonymity to quantify the privacy-preserving performance. We evaluated our solution on this test and corroborated that our learned semantic representations preserve speaker privacy, making it infeasible for state-of-the-art speaker identification models to link speakers between anonymized sets (linkability) or recognize the original identity of an anonymized sample (singling out). We showed the correlation of the proposed test with human perception by conducting an extra perceptual evaluation, where raters were unable to identify the original identity of the semantic reconstructed speech.

We have shown the trade-off between obfuscating speaker-identifiable traits and preserving useful, semantically rich information like prosody, sentiment, or emphasis while maintaining the content information. The more paralinguistic information is retained in the semantic representations, the more prone they are to breaking their privacy-preserving capabilities. Indeed, the way someone speaks is closely related to their identity. Additional research would benefit from loosening this tension, capturing further semantic paralinguistics without increasing the privacy risk.

## 6 ETHICAL STATEMENT AND RESPONSIBLE AI

The development of semantic privacy-preserving representations is motivated by the need to enable the widespread adoption of secure, speech-aware LLM-based models that do not compromise individual privacy. Ensuring this is crucial for upholding the principles of Responsible AI and fostering the public's and users' trust in generative speech technologies. Even given enough theoretical modeling capacity, the capabilities of a neural model are bottlenecked by the information encoded in its training targets. Consequently, models trained on USC semantic targets will generate expressive and natural speech that cannot be directly attributed to any specific individual. Moreover, explicit identity conditioning needs to be provided to complete the remaining speaker information for generating natural speech. Promising early results of speech disentanglement for Text-to-Speech (TTS) are shown in Appendix A, where we inject content and paralinguistics from privacy-preserving semantic representations and have control of the output identity through external conditioning.

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

# A  VOICE CONVERSION THROUGH SEMANTIC PARTIAL-TEACHER-FORCING

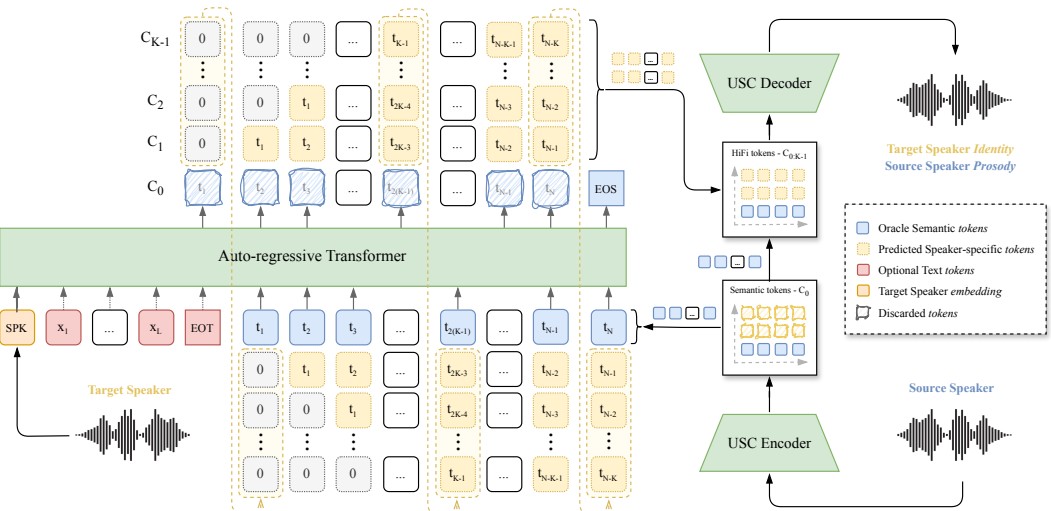

Figure 4: Voice Conversion pipeline for a LLM-based TTS model trained to predict USC representations. Semantics is extracted from source speaker (blue) and identity from target speaker (yellow).

To show the flexibility of disentangled USC representations, we trained an early LLM-based TTS model capable of predicting the full set of high-fidelity tokens from a speaker reference and input text. Our model builds upon the work by Łajszczak et al. (2024), with modifications to support multi-codebook prediction. The model autoregressively predicts all $K$ residual tiers of the USC, $C_{0:K-1}$, in a delayed pattern approach (Copet et al., 2023) and then generates high-fidelity speech through the USC decoder. A more extensive evaluation of this TTS model is left for future work.

While trained solely on the TTS task, the disentanglement of USC codes enables faithful voice conversion (VC) capabilities within the model. Figure 4 illustrates the inference approach of the trained TTS model for the VC process. The inference process involves teacher-forcing the disentangled semantic token $C_0$ from a source speaker. Consequently, during inference, the autoregressive prediction only generates the additional speaker-specific representations from the reference while preserving the content and paralinguistic information of the source speaker. We refer to this method as partial-teacher-forcing (PTF). Its effectiveness has been confirmed by informal listening tests through combining different source and reference speakers. Text is not even required for VC through PTF, and the model generates clear speech, showcasing the high content information encoded in the semantic codebook. Figure 5 illustrates one example, in which the duration and intonation from a male source speaker are preserved, but the generated female speech shows a converted higher pitch.

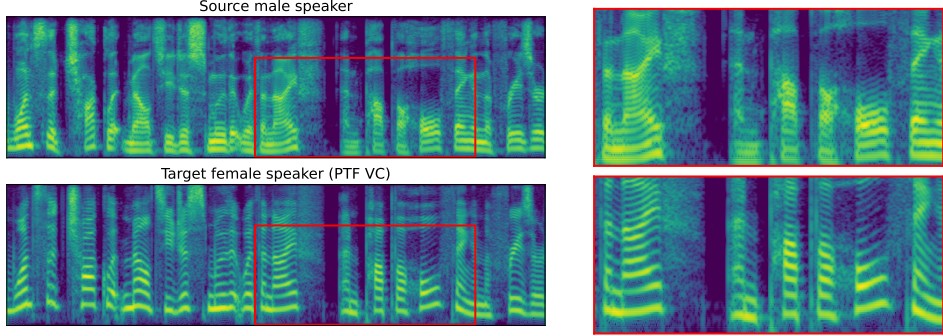

Figure 5: Mel-Sepctrogram visualization of VC capabilities through semantic $C_0$ partial-teacher forcing (PTF) without text: a source low-pitched speech converted to a target high-pitch speaker.

## B    MODEL ARCHITECTURE DETAILS

**Encoder**: The model is built as a sequential series of a pre-convolutional layer, a set of down-sampling blocks, and a post-convolutional layer. The pre-convolutional layer is a 1D convolution with a kernel size of 7 that maps the single-channel audio input into a higher-dimensional representation of size 64, which serves as the hidden dimension for the encoder. Then, the obtained latent representation is downsampled by passing through 4 different downsampling blocks. Each downsampling block comprises two components: (*i*) three residual blocks in sequence, where each residual block consists of a dilated 1D convolutional layer with a kernel size of 7, followed by a regular 1D convolutional layer with a kernel size of 1. The output of each convolutional layer is passed through a log-scale snake-beta activation function. The three residual blocks have dilations of (1, 3, 9) respectively. (*ii*) a strided 1D convolutional layer with kernel size double the amount of stride, for downsampling the output of the residual blocks. This strided convolutional layer doubles the amount input channels in its output. We set the downsampling strides for the 4 encoder down-sampling blocks to (2, 2, 4, 5, 8), leading to a total downsampling factor of $640\times$. Finally, a post 1D convolution with a kernel size of 3 is applied to the output of the last encoder downsampling block to project the encoded latent to a 768-dimensional latent.

**Decoder**: The decoder mirrors the encoder structure. The pre-convolutional layer has a kernel size of 3 and maps the 768-dimensional encoded latent to the decoder hidden dimension, which is set to 1536. Instead of 4 downsampling blocks, the decoder uses 4 upsampling blocks built through a nearest-neighbor upsampling followed by a 1D convolution. The upsampling rate is given by the stride factor of the upsampling block, and the following convolution has a kernel size of two times the rate factor. Upsampling layers divide by 2 the number of channels in the output. We set the upsampling rates for the 4 blocks to (8, 5, 4, 2, 2) respectively. Finally, the post-convolution layer maps the hidden dimension to the single waveform dimension.

Note that this decoder configuration reconstructs the same input waveform at 16 kHz. For training the up-sampler 24kHz decoder, we set the 5 upsampling layers with rates (8, 5, 4, 3, 2) for an upsampling factor of $940\times$ to upsample an encoded 16 kHz waveform to 24 kHz.

**RVQ**: We used 6 layers of RVQ with 16,384 tokens for the first codebook ($C_0$) and 1024 for the remaining residuals. We used L2-normalized codes, which means that the closest codebook entry is searched through the cosine distance of normalized latents. We also employ factorized VQ at each residual quantizer. Factorized VQ projects the input latent from a high-dimensional space $D$ to a denser space of dimensionality $M < D$ before quantizing through a learnable projection $W_{in} \in \mathbb{R}^{D \times M}$. Then, the codebook is learned in this low-dimensional dense space. After the quantization step, a learnable upsample projection $W_{out} \in \mathbb{R}^{M \times D}$ up-samples back the quantized latent to the original dimensionality. The original dimensionality is set to $D = 768$ which is the encoder hidden dimension, and the low-dimensional lookup embedding size is set to $M = 8$.

**Discriminators**: We use two types of discriminators defined in Section 3.6: MPD and MB-MRSD.

For the MPD, we use 5 identical period discriminators for each period. We set the periods to (2, 3, 5, 7, 11). Each period discriminator is a set of four 2D strided convolutional layers and a final 2D convolutional layer. Each strided convolutional layer has a kernel size of $3 \times 1$ and a stride of $(3, 1)$. The number of output channels of each strided convolutional layer is (32, 128, 512, 1024). The final convolutional $1 \times 1$ layer has the same input and output channels, 1024.

For MB-MRSD, we use 3 identical resolution discriminators for different STFT parameters. For each resolution discriminator, we set n_fft $= (2048, 1024, 512)$ respectively, with the hop length being a fourth part of n_fft and the window length being equal to n_fft. Each resolution discriminator is a set of four strided 2D convolutional layers and a final 2D convolutional layer. Each strided convolutional layer has a kernel size of $3 \times 9$ and a stride of $(1, 2)$. The number of output channels of each strided convolutional layer is 32, the same as the final $1 \times 1$ convolutional layer. For the multi-band, we multiply each resolution discriminators for each of the 5 sub-bands, ending up with 15 different discriminators. Each one is trained for a specific sub-band based on the percentage of STFT frequencies to take. We ranged our bands as 0-10%, 10-25%, 25-50%, 50-75%, and 75-100%.

**Speaker classifier**: It builds on a 4-layer encoder transformer architecture with a hidden size of 768 and 4 attention heads. The classifier layer returns the pooler output and projects it to the number of labeled speakers to apply the speaker classification loss.

## C  DERIVATION OF GENERAL RVQ FORMULATION

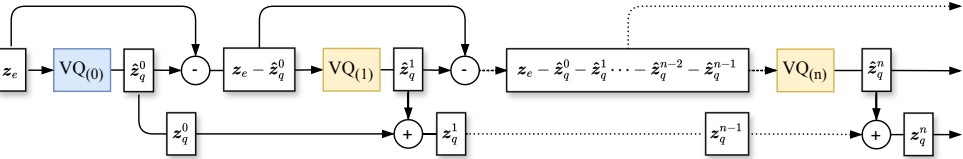

Figure 6: General RVQ architecture diagram for an arbitrary number of $n$ residual quantizers.

Looking at the RVQ diagram on Figure 6, for an arbitrary number $n$ of residual quantizers, the quantized residual representation $z_q^n$ of the input $z_e$ is recursively defined as:

$$z_q^n = z_q^{n-1} + \hat{z}_q^n \tag{6}$$

where $\hat{z}_q^n$ is the quantized error of the previous quantization step. By design of the residual vector quantization (RVQ) process, $\hat{z}_q^n$ can be expressed as the quantization of the residual error between $z_e$ and the cumulative quantized errors from all previous $n - 1$ quantization steps, i.e.,

$$z_q^n = z_q^{n-1} + \text{VQ}_{(n)}(z_e - \hat{z}_q^0 - \hat{z}_q^1 \cdots - \hat{z}_q^{n-2} - \hat{z}_q^{n-1}) \tag{7}$$

Substituting the expressions $\hat{z}_q^n = z_q^n - z_q^{n-1}$ (Equation 6) and $\hat{z}_q^0 = z_q^0$ (by RVQ design) into Equation 7, we obtain:

$$z_q^n = z_q^{n-1} + \text{VQ}_{(n)}(z_e - z_q^0 - (z_q^1 - z_q^0) - (z_q^2 - z_q^1) \cdots - (z_q^{n-2} - z_q^{n-3}) - (z_q^{n-1} - z_q^{n-2}))$$

And simplifying the resulting expression by canceling out terms that sum to zero:

$$z_q^n = z_q^{n-1} + \text{VQ}_{(n)}(z_e - \cancel{z_q^0} - \cancel{z_q^1} + \cancel{z_q^0} - \cancel{z_q^2} + \cancel{z_q^1} \cdots - \cancel{z_q^{n-2}} + \cancel{z_q^{n-3}} - z_q^{n-1} + \cancel{z_q^{n-2}}) \implies$$

$$\implies z_q^n = z_q^{n-1} + \text{VQ}_{(n)}(z_e - z_q^{n-1}) \tag{8}$$

Therefore, Equation 8 provides a recursive formulation for computing the quantized residual representation $z_q^n$ at any desired level $n$, given the initial quantizer input $z_e$ and the previous quantized residual representation $z_q^{n-1}$. However, by exploiting the recursive nature of the presented equation, we can derive a general expression for $z_q^n$ in terms of the initial quantized representation $z_q^0$ and the sum of quantized residual errors from all $n$ quantization steps:

$$z_q^n = z_q^0 + \sum_{i=1}^{n} \text{VQ}_{(i)}(z_e - z_q^{i-1}) \tag{9}$$

which is the RVQ Equation we presented in Section 3.1.

## D Upsampling 24 kHz decoder

A substantial amount of the speech data exists only at a sampling rate of 16 kHz and/or in the MP3 encoded format, where high-frequency information is heavily compressed for format efficiency. Therefore, USC is trained on 16 kHz input data with the goal of ensuring that speech representations of different sampling rates are mapped to the same token. In other words, the same speech sound from a high rate source should correspond to the same representation as the same sound from a low rate source. If the encoder operated at a high frequency input, the learned codebook would map different tokens for the same speech sound. Therefore, the training process of the codec model would result in a codebook that uses part of its capacity to learn frequency-based representations rather than semantic-based ones, independent of the sampling rate of the input speech.

Speech is a periodic signal in which higher harmonics can be extrapolated from low-frequency information (Liu et al., 2022). Following the two-step process proposed by Wu et al. (2023), we have trained a final 24 kHz decoder with a frozen 16 kHz encoder and RVQ component. As noted by Kumar et al. (2023), careful selection of full-band data is important to achieve artifact-free 24 kHz waveform reconstruction. We introduced a set of energy-based filters and careful data-processing to ensure that, from all the pre-training data used, we only use full-band 24 kHz waveforms to train the upsampled decoder. Details on the upsampling decoder architecture are found in Appendix B.

## E Training hyperparamters

We train USC with the balancing loss presented in Section 4.1 for 1M steps. In the proposed GAN-based training, we optimize the generator network (comprised of the speech codec, the speaker classifier, and the semantic distillation) and the discriminators through the Adam Optimizer. We set the learning rate to 0.0001, with $\beta_1 = 0.8$ and $\beta_2 = 0.99$. We use a warmup learning rate decay strategy that sets the maximum learning rate to $10^{-4}$ and the minimum learning rate to $10^{-7}$. The warmup stage lasts 10K steps. We clip the gradients norm to 10.

We train USC with 3-second segments, which corresponds to approximately 75 discretized latents per sample. We use a batch size of 8 per GPU, and we train our model on 4 nodes, each with 8 NVIDIA A100 GPUs, thus resulting in an effective batch size of 256. We use the same hyperparameters when training the 24 kHz decoder (frozen Encoder and RVQ) for 2.5M steps on only perceptual and reconstruction losses.

## F Bit-rate computation

The bandwidth (BW) of a neural audio codec is given by the sampling rate at which the codec operates, the number of residual codebooks, and the amount of downsampling factor applied to the input waveform (Table 3). The downsampling factor is given by the product of all the stride values in the audio codec. For a variable number of $n$ residual codebooks, the bandwidth is given by:

$$\text{BW } C_{0:n} \text{ (kbps)} = \frac{\text{Sample rate (kHz)}}{\text{Factor}} \times \sum_{i=0}^{n} \lceil \log_2(\#C_i) \rceil \tag{10}$$

Table 3: Architecture hypermeters used for calculating the bandwith of each model and the computed bit-rate for both semantic $C_0$ and high-fidelity $C_{0:K-1}$ reconstruction.

| Model | Sample rate | Strides | Factor | #$C_0$ | #$C_{1:K-1}$ | #K | BW $C_0$ | BW $C_{0:K-1}$ |
|---|---|---|---|---|---|---|---|---|
| EnCodec | 24 kHz | $(2, 4, 5, 8)$ | $320\times$ | 1024 | 1024 | 8 | 0.75 kpbs | 6.00 kpbs |
| DAC | 44.1 kHz | $(2, 4, 8, 8)$ | $512\times$ | 1024 | 1024 | 9 | 0.86 kpbs | 7.75 kpbs |
| SpeechTokenizer | 16 kHz | $(2, 4, 5, 8)$ | $320\times$ | 1024 | 1024 | 8 | 0.50 kpbs | 4.00 kpbs |
| FaCodec | 16 kHz | $(2, 4, 5, 5)$ | $200\times$ | $2 \times 1024$ | 1024 | 5 | 1.60 kpbs | 4.80 kpbs |
| USC (Encoder) | 16 kHz | $(2, 2, 4, 5, 8)$ | $640\times$ | 16,384 | 1024 | 6 | 0.35 kbps | 1.60 kbps |
| USC (Decoder) | 24 kHz | $(8, 5, 4, 3, 2)$ | $960\times$ | | | | | |

## G  PITCH CONTOUR FOR SEMANTIC WAVEFORM RECONSTRUCTION

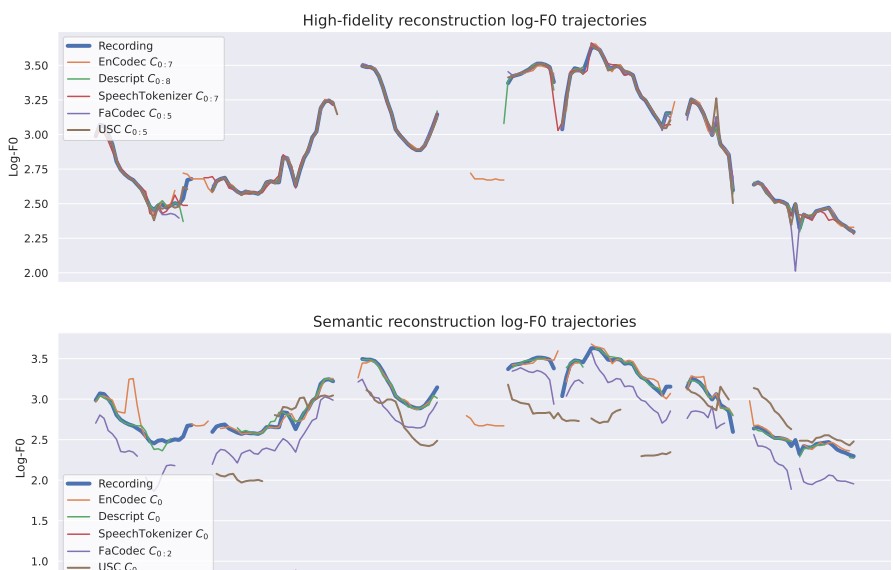

Figure 7: Comparison of log-F0 contour for high-fidelity and semantic waveform reconstruction.

Fundamental frequency (F0 or pitch), can be considered a type of prosody measurement. We present three objective metrics in Table 4. Figure 7 presents the log-F0 contour for both high-fidelity reconstruction (using all the RVQ codebooks) and semantic reconstruction (using only the first codebook).

For high-fidelity reconstruction, all the models are able to re-generate the original pitch contour, with some slight artifacts that are negligible for the study of speaker disentanglement.

For semantic reconstruction, EnCodec and DAC are the two models that better reconstruct the original pitch of the waveform from the first codebook as they don't apply any speaker anonymization technique. However, while being able to reconstruct the original F0 trajectory, the content preservation for these two models is poor, indicating that the first codebook has learned to encode acoustic information before speech content. For SpeechTokenizer, it is completely the opposite. While it reports high WER/CER metrics, the semantic reconstruction pitch trajectory is completely null, showing a constant robotic pitch. This observation is evident in the correlation metrics in Table 4, and aligns with the sentiment CCC metric in Table 1. FaCodec, however, is able to reconstruct the most correlated pitch trajectory without sacrificing the WER metric. However, this comes at the cost of needing the highest amount of bit-rate of all the baselines and two different quantizers for content a prosody. USC is able to maintain the pitch trajectory of the speech while maintaining competent WER/CER metrics. However, this trajectory doesn't correspond with the original recording's octave of the pitch, which showcases the speech privacy-preserving capabilities of the semantic representations, generating different identities within the same utterance.

Table 4: F0 Spearsman Correlation Coefficient (SCC), Pearson Correlation Coefficient (PCC) and root mean squared error (RMSE) metrics across all the reported systems.

| Model | *High-Fidelity Reconstruction* | | | *Semantic Reconstruction* | | |
| --- | --- | --- | --- | --- | --- | --- |
| | SCC ↑ | PCC ↑ | RMSE ↓ | SCC ↑ | PCC ↑ | RMSE ↓ |
| EnCodec | 0.891 | 0.901 | 0.133 | 0.641 | 0.632 | **0.324** |
| DAC | **0.962** | **0.968** | **0.051** | 0.728 | 0.701 | **0.311** |
| SpeechTokenizer | **0.957** | **0.963** | 0.075 | 0.118 | 0.117 | 1.285 |
| FaCodec | **0.961** | **0.967** | 0.069 | **0.815** | **0.790** | 0.353 |
| USC | **0.959** | **0.963** | 0.071 | 0.526 | 0.486 | 0.430 |

## H  RANDOM GUESSING DISTRIBUTION ANALYSIS

In the privacy-preserving test, random guessing through indistinguishable samples would generate ranks that follow a uniform distribution $\bar{r}_s \sim \mathcal{U}(1, N)$ over the possible $N$ rank options. Therefore, the mean $\mu$ and variance $\sigma^2$ of a uniformly distributed variable are given by:

$$\mu = \mathbb{E}[\bar{r}_s] = \sum_{i=1}^{N} \frac{1}{N} i = \cdots = \frac{N+1}{2} \tag{11}$$

$$\sigma^2 = \mathbb{E}\left[(\bar{r}_s - \mathbb{E}[\bar{r}_s])^2\right] = \sum_{i=1}^{N} \left(i - \frac{N+1}{2}\right)^2 \frac{1}{N} = \cdots = \frac{(N-1)^2}{12} \tag{12}$$

According to the Central Limit Theorem (CLT), a sum of a large number of independent and identically distributed random variables approaches a normal distribution. The mean of this normal distribution is equal to the mean of the individual random variables, and the variance is equal to the variance of the individual random variables divided by the number of independent tests. Therefore, for $L$ different privacy-preserving tests:

$$\mu_L = \mu = \frac{N+1}{2}, \quad \sigma_L^2 = \frac{\sigma^2}{L} = \frac{(N-1)^2}{12L} \tag{13}$$

Given the normal rank distribution of random guessing for $L$ test we can compute the 50th percentile (p50) and the 1st percentile (p1) of it. The p50 corresponds to the median of the normal distribution, which for the presented distribution is directly $\mu_L$. For the p1, we first get the $z$-score for the first percentile from a standard normal statistical distribution table. The z-score for the first percentile, $z_1$, is approximately $-2.326348$. Then we obtain the value of the first percentile using the following equation:

$$\text{p1} = \mu_L + z_1 \sqrt{\sigma_L^2} \tag{14}$$

Now, for the proposed privacy-preserving test detailed in section 4.3, we use $N = 7947$ different speakers for $L = 100$ different tests. Using the Equation 13, the mean of the distribution for random guessing is $\mu_L = 3987.50$, which is also the 50th percentile of the distribution. The variance $\sigma_L^2 = 52973.94$ and, substituting this value with the 1st percentile $z$-score in the Equation 14, we get a value of p1 $= 3452.06$. These two values are the ones reported as ceilings for the privacy-preserving metrics in Table 2.

# I    FURTHER SEMANTIC MEL-SPECTROGRAM EXAMPLES

In the visual analysis of the presented mel-spectrograms, the F0, or pitch, is evident as the lowest energetic horizontal structure. Parallel structures represent harmonics, occurring at integer multiples of the F0. The spacing and intensity of these harmonics contribute significantly influence the overall pitch of the generated voice and, consequently, the perceived identity. The temporal consistency and variability of the F0 and harmonic patterns indicate specific voice attributes, such as intonation and emphasis, which serve as differentiating paralinguistic features.

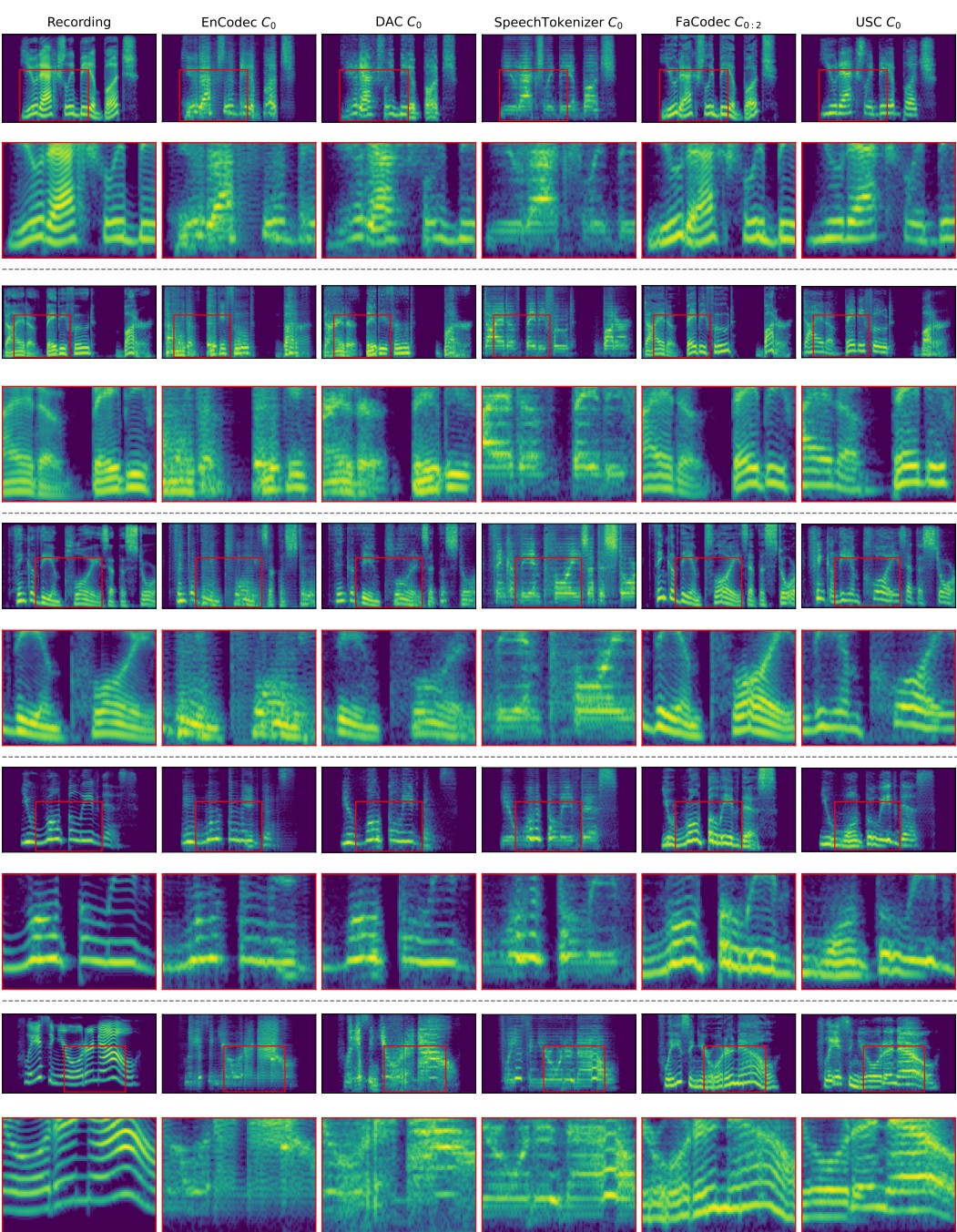

Figure 8: 5 semantic reconstruction mel-spectrograms of different speakers from the evaluation set.

## J   HUMAN PRIVACY-PRESERVING EVALUATION TESTCASE EXAMPLE

Figure 9 presents a test case from our perceptual privacy-preserving human test. In it, the participant is presented with a reference (original) audio sample and two semantic reconstructions of another speech sample. One of the two candidate samples corresponds to the same speaker as the reference. The participant is instructed to listen to all three samples and then move a slider to indicate which of the two semantic waveform candidates most closely resembles the reference sample in terms of speaker similarity. Only one option can be selected, and the 'Submit' button becomes available after making a choice between the two candidates.

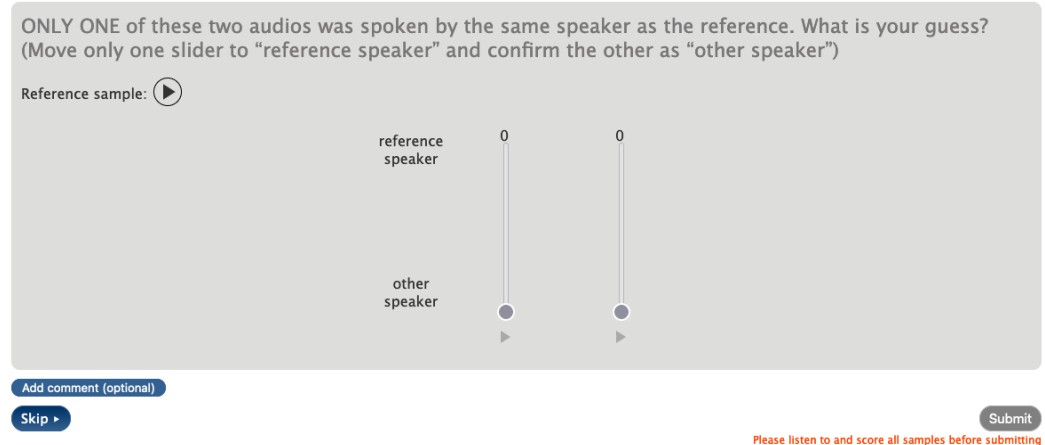

Figure 9: Random test-case screenshot from the subjective privacy-preserving evaluation.

