# OpenReview forum: "Universal Semantic Disentangled Privacy-preserving Speech Representation Learning"
_ICLR.cc/2025/Conference — Submitted to ICLR 2025_

### Official Review · Reviewer_YwDt · 2024-10-31

**Soundness:** 3
**Presentation:** 4
**Contribution:** 4
**Rating:** 6
**Confidence:** 5

**Summary:**

The paper addresses the important question of privacy in training LLMs and proposes a method to anonymize the input to training. It does so by applying an anonymizing autoencoder structure where parts of the bottleneck is fed to the LLM instead of the original signal.
It is a large paper with many strong aspects, but unfortunately, there are also some critical weaknesses. The papers presents a comprehensive and useful review of related literature and describes the proposed approach clearly.

**Strengths:**

- Addresses a substantial problem of privacy in LLM training.
- Very good description of background
- Thorough and detailed presentation of the proposed model.

**Weaknesses:**

- There are so many ways in which private information can leak through that it is important to never claim that privacy is fully preserved for the speech signal, but only for a selection of attributes.
- STOI is a very old metric for speech intelligibility and does not correlate well with quality. The high-fidelity scores are especially clearly saturated. MCD is even older and equally sub-optimal. The quality comparison is thus not trustworthy. AFAIK the community currently prefers VISQOL and I would advise switching to that.
- Claiming high quality speech reconstruction with only objective metrics is not acceptable. Even with state-of-the-art metrics like Visqol, we can always find counter-examples where subjective preference is the opposite of objective metrics. Claims of high-quality reconstruction must, therefore, be accompanied by subjective listening tests.
- None of the metrics include an analysis of statistical significance.
- The description of a key piece of the privacy measure, the SIM metric, is very short and hard to find. In comparison, sections 3.5 and 4.3 receive a lot of real estate. Please fix this imbalance by describing SIM in more detail.

**Questions:**

- How does your method rank against other speech disentanglement approaches? Now, you were comparing only to codecs.

---

> ### Author Response · Authors · 2024-11-21
>
> Thank you very much for the review and suggestions. We address your main question and described weaknesses below:
>
> > How does your method rank against other speech disentanglement approaches? Now, you were comparing only to codecs.
>
> We have added a new baseline to the paper, FACodec, which evaluates another disentanglement approach. Most of our tests are carried out on reconstruction basics; therefore, we compare our approach against methods that discretize speech and are able to generate waveforms without external models or training. This is why we focus our evaluation on codecs.
>
> > There are so many ways in which private information can leak through that it is important to never claim that privacy is fully preserved for the speech signal, but only for a selection of attributes.
>
> We agree. We have rephrased **line 511** to claim that our representations preserve speaker privacy for non-prosodical attributes, which are also characteristics that could enable speaker identification, as noted in line 520.
>
> > The high-fidelity scores are especially clearly saturated. MCD is even older and equally sub-optimal.
>
> We have added the extra PESQ metric as an additional quality metric. Thanks for the suggestion.
>
> > Claiming high quality speech reconstruction with only objective metrics is not acceptable. Even with state-of-the-art metrics like Visqol, we can always find counter-examples where subjective preference is the opposite of objective metrics. Claims of high-quality reconstruction must, therefore, be accompanied by subjective listening tests.
>
> Thanks for the feedback. High-quality speech reconstruction should be accompanied by subjective tests for claiming so. Given the scale and costs of these evaluations, we believe it is more appropriate to report and evaluate the privacy-preserving characteristics of our representations (what we aim to show) and rely on objective metrics and listening examples for the high-fidelity reconstruction.
>
> > None of the metrics include an analysis of statistical significance.
>
> We highlight the best metric in our objective table with statistical significance at a $p_{\text{value}} > 0.05$. It is true that we didn't report statistically significant differences across non-best systems to avoid overbloating the objective metric table in the given space.
>
> > The description of a key piece of the privacy measure, the SIM metric, is very short and hard to find. In comparison, sections 3.5 and 4.3 receive a lot of real estate. Please fix this imbalance by describing SIM in more detail.
>
> We have improved the SIM metric description and overall Section 4.2 for clearer definitions. We still believe that Section 4.3 needs a detailed explanation, as we consider it a new test definition for privacy preservation.

---

> > ### Comment · Reviewer_YwDt · 2024-11-22
> > **Evaluation of output quality was not sufficiently improved in revision**
> >
> > - Adding PESQ, another antiquated objective quality metric, does not solve the problem. VISQOL, at the minimum, would be required to have a barely acceptable quality metric. Subjective listening tests would be the gold standard.
> > - Focusing on privacy-preserving metrics is essential, but completely dismissing subjective evaluation is folly. Without subjective listening, there is no guarantee that the output sound quality is usable. I'm not saying it is, but subjective quality *could* be horrific, and we would never notice.
> > - I notice that the other reviewers have given detailed and important feedback on the relationship with the VoicePrivacy challenge, so I don't need to do so here. I just want to say that I noticed that those reviewers are obviously experts, so their feedback has to be taken seriously.

---

> > > ### Author Response · Authors · 2024-11-22
> > >
> > > Thanks again for your suggestion, and apologies for not addressing your comment on the ViSQOL metric earlier. We have decided to replace the MCD metric with the ViSQOL metric as you suggested, and changed the narrative accordingly.
> > >
> > > Given the large-scale comparison of systems and samples, we have prepared a MUSHRA test for high-fidelity reconstruction across all the baselines. We will update and notify once we process the results and add them to the evaluation.

---

### Official Review · Reviewer_vjTT · 2024-11-01

**Soundness:** 2
**Presentation:** 3
**Contribution:** 2
**Rating:** 5
**Confidence:** 4

**Summary:**

This paper describes the Universal Speech Codec (USC), which disentangles speech into global characteristics and time-varying content in a manner useful for speaker anonymization. The authors build upon the DAC audio codec, adding techniques such as reversing speaker gradients, distilling semantic representations from a pre-trained SSL model, and local differential privacy. Results demonstrate comparable reconstruction quality as preceding codecs, and privacy evaluations demonstrate performance not clearly exceeding prior methods.

**Strengths:**

The privacy evaluations described in section 4.3 are designed and described well.

I like the framing of disentanglement for privacy applications. I think there are some genuinely useful applications.

**Weaknesses:**

The authors point out that USC "generat[es] different identities across the same utterance." Wouldn't this be a clear give-away that the audio is generated if the speaker is changing throughout the utterance? Is there a use case in which having audio with a varying speaker identity is desirable?

While the LDP is ablated, the other proposed methods (e.g., speaker reversal and semantic distillation) are not sufficiently ablated.

The subjective evaluation of section 4.4 should be performed using a baseline method such as Speech Tokenizer, as well as original audio.

The voice conversion experiment is incomplete. A privacy-preserving speech synthesizer should be able to perform voice conversion, but the single example provided in Appendix A (without audio) is not sufficient to demonstrate voice conversion at any threshold of quality. If a listening test was performed for VC, why were results not included? And what was the baseline of those listening tests?

It doesn't seem particularly fair to compare bit rates with DAC when DAC models 44.1 kHz audio and the proposed model models 24 kHz audio.

**Questions:**

Can audio examples be included, so that we may compare your system to the baseline systems using our own ears?

---

> ### Author Response · Authors · 2024-11-21
>
> Thank you for your detailed review. Audio samples are included in the supplementary material as an HTML sample page. Please let us know if you encounter any issues accessing or listening to them.
>
> Regarding the questions you had in the weaknesses section:
>
> > Wouldn't this be a clear give-away that the audio is generated if the speaker is changing throughout the utterance? Is there a use case in which having audio with a varying speaker identity is desirable?
>
> Listening to the samples should clarify this point. When we mention that USC "generates different identities across the same utterance" we mean that reconstructing speech from C0s (which lack speaker-specific information) leads to the synthesis of different identities. This occurs because the decoder attempts to generate faithful human speech from representations that don't have an assigned identity, resulting in a random set of identities when reconstructing a semantic waveform.
>
> > While the LDP is ablated, the other proposed methods (e.g., speaker reversal and semantic distillation) are not sufficiently ablated.
>
> You are correct that we only ablated the LDP component. The other components were previously ablated in their original works. We chose to focus on ablating the LDP component as it is new to speech codecs in the literature. Evaluating it through a carefully designed test of the privacy-preserving capabilities is of most interest to the scientific community.
>
> > The subjective evaluation of section 4.4 should be performed using a baseline method such as Speech Tokenizer, as well as original audio.
>
> We primarily used the subjective evaluation to confirm that our objective privacy-preserving design is effective. As noted in **line 365**, subjective human tests are not the best metric for comparing speaker identities. Even if we added SpeechTokenizer to the test, we wouldn't expect to achieve better than a 0.5% accuracy (essentially coin-tossing), which is what we achieve with the proposed method.
>
>  > If a listening test was performed for VC, why were results not included? And what was the baseline of those listening tests?
>
> We have provided VC samples on the sample page. Please inform us if you have any problems accessing them. As stated in line 784 of Appendix A, we plan to evaluate the developed TTS model (with VC capabilities) more thoroughly in future work. This paper's evaluation methodology primarily focus on the privacy evaluation designed for the learned representation, with Appendix A demonstrating a downstream utility example of the proposed disentanglement.
>
> > It doesn't seem particularly fair to compare bit rates with DAC when DAC models 44.1 kHz audio and the proposed model models 24 kHz audio.
>
> We considered using the 24 kHz variant of DAC for our evaluation. However, that variant had 32 codebooks (24 kbps) and was not evaluated in their original work. Moreover, following the formula presented in Appendix F, the 24 kHz variant of DAC has a bit rate of 0.75 kbps for its first codebook (similar to EnCodec), which is still higher than what we propose for our 24 kHz variant.

---

> ### Comment · Reviewer_vjTT · 2024-11-23
>
> > The other components were previously ablated in their original works.
>
> I stand by the value of more ablations. While those components (speaker reversal and semantic distillation) have been ablated elsewhere, the overall system it was ablated with is unique. As well, if ablations show all of the changes relative to a known baseline, it becomes much easier to discern the value of each component.
>
> > As noted in line 365, subjective human tests are not the best metric for comparing speaker identities.
>
> You cannot use this citation to make sweeping generalizations discrediting well-designed subjective evaluations. That citation you are mentioning in line 365 uses non-native speakers as participants for an A/B test. You should be using native speakers and (as another reviewer pointed out) ABX tests. MUSHRA tests are also useful if you have more than two conditions to compare to a reference.
>
> > We have provided VC samples on the sample page
>
> That's good that we can now listen to samples. But if you are not going to provide details of the listening test, you should consider omitting this line from the voice conversion section of the paper (lines 791-792):
>
> >Its effectiveness has been confirmed by informal listening tests through combining different source and reference speakers.
>
> But even better would be to conduct a well-designed subjective evaluation.
>
> I maintain my current review score of a 5.

---

### Official Review · Reviewer_JxJM · 2024-11-03

**Soundness:** 2
**Presentation:** 3
**Contribution:** 2
**Rating:** 5
**Confidence:** 4

**Summary:**

This paper introduced a new kind of speech representation learning for the privacy-preserving neural codec. The proposed method disentangled the speech information into two parts: semantic information containing both the content and paralinguistic parts of the speech and the speaker information. Several experiments were conducted to show the effectiveness of the proposed method in aspects of both the reconstruction capability and the privacy-preserving properties of the proposed codec by comparing the proposed method with existing codec systems. An additional experiment of training a voice conversion system with the proposed codec was performed to demonstrate the efficacy of the proposed codec in downstream applications.

**Strengths:**

1. The paper looks into the privacy-preserving aspects of the neural speech codec, which is important but not the main focus of previous studies.

2. Details on the model architecture and training are provided, enhancing the reproducibility of this paper.
3. The authors provided new metrics for privacy-preserving evaluation, which can benefit future research.

**Weaknesses:**

1. **Baseline missing**: In the related works, FACodec [1] is mentioned to be disentangled for some speech attributes. However, FACodec was not included as one of the baselines. It will be better to include this baseline for comparison.

2. **Overall Performances**: While the authors reported several metrics, the proposed method did not have much advantage over the baselines or even seemed to be inferior. Though the authors provided some qualitative examples, trying to explain this phenomenon, I think some quantitative measurements are still necessary to convince the readers that the proposed method is indeed better. Hence, some additional experiments are needed in my opinion.

3. **Missing Experiments**: The authors did not have a quantitative evaluation of the trained voice conversion model. Thus, it would be hard to convince the reader that the proposed codec and the Partial-Teacher-Forcing (PTF) techniques were really helpful. In addition, it seems like ABX tests for other baselines were absent in the current version of the paper, making the evaluation less consistent. The authors are suggested to include the tests.

4. **Clarity**: The authors provided two metrics for privacy-preserving evaluation. However, the definitions of these metrics were not easy to understand at first glance. If a visual demonstration can be included in the paper, the clarity will be largely improved.



[1] Ju, Zeqian, et al. "Naturalspeech 3: Zero-shot speech synthesis with factorized codec and diffusion models." arXiv preprint arXiv:2403.03100 (2024).

**Questions:**

My main concerns are listed in the weakness part. Here, I list some questions for the author as follows:

1. Why was FACodec not included as one of the baselines? As aforementioned, disentanglement was conducted for FACodec. Thus, it should be a good baseline to compare.

2. The authors mentioned some trade-offs between privacy-preserving aspects and some speech attributes like paralinguistic information. Is there any way to quantitatively evaluate this aspect? This is related to the second point of the weakness. I think the results so far are not sufficient to convince the readers that the proposed method is really better.

3. As mentioned in the weakness, I think ABX tests for other baselines are necessary. The authors can consider to include them.

4. Some in-house datasets were used for the experiments. Some explanations on how these data were collected will make the paper better if included even though you may not want to release them.

5. I suggest that more detailed explanations of the qualitative examples (mel-spectrograms) should be provided to make the paper easier to understand.

A minor issue is that some typos need to be fixed (e.g., i.e should be i.e. in Sec. 3.4). Proofreading is necessary for future versions of this paper.

---

> ### Author Response · Authors · 2024-11-21
>
> Thank you very much for the detail feedback and review of our work. Let us address you questions below:
>
> > Why was FACodec not included as one of the baselines? As aforementioned, disentanglement was conducted for FACodec. Thus, it should be a good baseline to compare.
>
> We appreciate your suggestion and have included FACodec in our baseline comprison. As FACodec requires a speaker embedding for synthesis we combined the averaged speaker identity across all evaluation speakers. This approach is now explained in **line 430**.
>
> > The authors mentioned some trade-offs between privacy-preserving aspects and some speech attributes like paralinguistic information. Is there any way to quantitatively evaluate this aspect? This is related to the second point of the weakness. I think the results so far are not sufficient to convince the readers that the proposed method is really better.
>
> We have attempted to isolate this evaluation using different metrics such as sentiment preservation and F0 correlation. This approach aligns with trends in the Voice Privacy Preserving challenge [1], as noted by reviewer Prj4, and we have included this work in our evaluations. Our method achieves better speaker similarity and better Sentiment/F0 correlation compared to SpeechTokenizer, where their sentiment preservation in semantic representations is minimal.
>
> We have included semantic waveform reconstructions in the supplementary material to illustrate the differences between semantic speech with and without sentiment information. Additionally, we showcase examples of Voice Conversion, demonstrating the preservation of the source's intonation and non-verbal sounds in the converted sample. We are reserving a detailed evaluation of the TTS and Voice Conversion model for future work, as we believe the privacy preservation analysis is the most significant and relevant part of the presented methodology.
>
> > As mentioned in the weakness, I think ABX tests for other baselines are necessary. The authors can consider to include them.
>
> Thanks for the suggestion. ABX testing for other baselines could be insightful. However, the primary purpose of the ABX test is to corroborate the objective privacy-preserving test. As cited in **line 369**, subjective human tests are not the optimal metric for comparing speaker identities. We demonstrate the lower ceiling of non-anonymization and our current method, which essentially results in a coin toss.
>
> > Some in-house datasets were used for the experiments. Some explanations on how these data were collected will make the paper better if included even though you may not want to release them.
>
> We have added some description on how the proprietary dataset was obtained.
>
> > I suggest that more detailed explanations of the qualitative examples (mel-spectrograms) should be provided to make the paper easier to understand.
>
> Thank you for this suggestion. While space is limited in the first 10 pages, we have included a reference and a brief explanation of understanding the mel-spectrograms attached in Appendix I.
>
> Thanks again for your review. We have also addressed some typos in the uploaded revision of the paper.
>
> ---
>
> [1] The VoicePrivacy 2024 Challenge Evaluation Plan

---

> ### Comment · Reviewer_JxJM · 2024-11-24
>
> > We appreciate your suggestion and have included FACodec in our baseline comprison. As FACodec requires a speaker embedding for synthesis we combined the averaged speaker identity across all evaluation speakers. This approach is now explained in line 430.
>
> Thanks for the revision. However, I still have concerns about the performance of the proposed method, as it does not show superior performances over the baselines on the quantitative metrics. Similar concerns are raised by other reviewers as well. The current results are not convincing to show that the proposed method is better than any other baseline in terms of quantitative performances.
>
> > We are reserving a detailed evaluation of the TTS and Voice Conversion model for future work, as we believe the privacy preservation analysis is the most significant and relevant part of the presented methodology.
>
> I believe that the evaluation of the TTS and VC models using the proposed codec will be essential to show the practical utility of the proposed method besides the privacy preservation properties. The privacy preservation characteristics are orthogonal to the utility of the codec, which means that a codec system can be good in terms of privacy preserving but poor for the real-world applications. Thus, I still suggest that the detailed evaluation will be necessary. This is also pointed out by other reviewers.
>
> Currently, I will maintain the score. But I am willing to adjust it if the paper is refined.

---

### Official Review · Reviewer_6QEh · 2024-11-07

**Soundness:** 3
**Presentation:** 2
**Contribution:** 3
**Rating:** 5
**Confidence:** 3

**Summary:**

The paper introduces a new discrete tokenization method which has a non-speaker related initial token (semantic/phonetic and paralinguistic information is covered by this single token) and the rest of the tokens (per frame) cover the rest of the acoustic information (including speaker id) that helps with reconstructing a perceptually equivalent version of the speech waveform.

The separate roles of each token is enforced by using a bunch of losses that encourage non-speaker information to be used in the first level token (z^0_q) and regular codec losses.

The discretization is done through RVQ for the acoustic tokens.

The speaker information is removed from the first token using an adversarial speaker-id loss and a "differential privacy noise adding" module and through semantic distillation which tries to predict Hubert encodings from the semantic encodings (obtained from semantic tokens).

**Strengths:**

The idea of the paper is interesting in the sense that separating semantic and acoustic information in a single tokenizer is useful for privacy preservation through anonymization. This is achieved through multiple losses in the paper.

The privacy preservation of the method is analyzed through interesting metrics of linkability, singling out and perceptual privacy evaluations through A/B/X tests. These seem to be good privacy preservation tests.

**Weaknesses:**

SpeechTokenizer removes all non-phonetic information since it uses Hubert distillation for the semantic tokens. This method tries to leave both phonetic and paralinguistic (prosody, sentiment, etc.) information in the initial token, so that we can reconstruct a waveform from the initial token that resembles the original speech but without giving away the original speaker, thus achieving anonymization as claimed in the paper. Still, additional utility of this model as compared to SpeechTokenizer seems limited since SpeechTokenizer also achieves anonymization, but at the expense of losing sentiment and prosody etc. I think the paper needs to make the point that the goal is to also keep sentiment information for example, but we do not see a test for that in the paper. In the anonymization tests, SpeechTokenizer shines, so probably we need another test that shows that the method introduced in the paper beats SpeechTokenizer. Maybe sentiment analysis?

The description of modules are not very precise. Section 3.2 is not clear in that we cannot tell how the z^0_q relates to F_i. Also F_i where i indexes the batch item does not seem to make sense since we probably want F_(s_i) where s_i is the speaker id (out of N) of batch item i out of n.

Section 3.4 second paragraph is not clear. It describes a mechanism to stop gradients, but does not specify how it is done. Is it automatically achieved through losses or not?

Similarly, Section 3.5 is not clear how the noise is added and what it is added to. The paper did not talk about projections in the quantization module at this point but Section 3.5 talks about projections. We only read about projections in the appendix. W_in in Figure 2 is only explained in the appendix. Also, differential privacy usually adds noise to gradients (If I recall correctly), so is it correct to call what is done differential privacy or is it something else?

Equation (4) gives a bunch of losses with weights. It seems hard to tune those weights on different datasets since there are too many weights. How did you tune them?

**Questions:**

1. Please address points mentioned in the weaknesses.

**Details Of Ethics Concerns:**

N/A.

---

> ### Author Response · Authors · 2024-11-21
>
> Thanks for the comments and valuable feedback. Let me address the exposed weaknesses per points:
>
> > Still, additional utility of this model as compared to SpeechTokenizer seems limited since SpeechTokenizer also achieves anonymization, but at the expense of losing sentiment and prosody etc. I think the paper needs to make the point that the goal is to also keep sentiment information for example, but we do not see a test for that in the paper.
>
> > In the anonymization tests, SpeechTokenizer shines, so probably we need another test that shows that the method introduced in the paper beats SpeechTokenizer. Maybe sentiment analysis?
>
> As you correctly pointed out, the main difference between our approach and SpeechTokenizer is that our method encapuslates more paralinguistic speech information in the semantic representations. We motivate that capturing this information is crucial for a better representation of speech, as semantic representations without paralinguistics becomes a closer representation to text than speech. This idea is being presented in **line 053**, and it is indeed one of the main goal and novelties of our research.
>
> For testing it, we show two different metrics: CCC Sentiment metric and F0 SCC correlation (with extended analysis in Appendix G). These are the standard sentiment metrics used to measure the amount of emotion encoded in generated speech. We show that the methods that encapsulate more sentiment are also the methods that don't anonymize their representations. USC finds a sweet spot, achieving good privacy-preserving metrics while maintaining speech sentiment in its semantic representations, all while being the lowest bit-rate codec among the baselines. Again, USC outperforms SpeechTokenizer since, basically, SpeechTokenizer doesn’t encode anything related to the sentiment in their semantic targets.
>
> As an extra, we demonstrate paralinguistic retention through a downstream Voice Conversion task in the Appendix A (with attached samples in the supplementary material).IIn the samples, it can be heard that paralinguistic information, such as intonation, pacing, and non-verbal sounds, is preserved and transplanted from the source speaker to the target speaker.
>
> > The description of modules are not very precise. Section 3.2 is not clear in that we cannot tell how the z^0_q relates to F_i. Also F_i where i indexes the batch item does not seem to make sense since we probably want F_(s_i) where s_i is the speaker id (out of N) of batch item i out of n.
>
> The equation does overload some notation and we apologise for the confusion. But as written $F_i$ is the logit for $speaker_i$ which for brevity is assumed to overload the speaker $i$ in the batch. As shown in the diagram, the logit is obtained from a speaker classifier which has num_speaker label space and where Lams is applied.
>
> > Section 3.4 second paragraph is not clear. It describes a mechanism to stop gradients, but does not specify how it is done. Is it automatically achieved through losses or not?
>
> We have re-phrased the section 3.4 to make it more clear. At training time, when the first quantizer is chosen (c0) as per dropout mechanism, we stop the gradients to the encoder. Losses remain the same. This approach ensures that only semantic quantizer and the decoder are optimized to generate the most faithful speech reconstruction from the C0s alone.
>
> > Similarly, Section 3.5 is not clear how the noise is added and what it is added to. The paper did not talk about projections in the quantization module at this point but Section 3.5 talks about projections. We only read about projections in the appendix.
>
> Thanks for notting this. We define the type of VQ used in line 195, where we specify that we use factorized VQ, defined in the work by [2]. These variant of VQ includes the input $W_{in}$ and output $W_{out}$ projections. In any case, we have extended **line 196** to explain the definition of factorized codebooks, including the mentioned projections.
>
> > Also, differential privacy usually adds noise to gradients (If I recall correctly), so is it correct to call what is done differential privacy or is it something else?
>
> The local differential privacy we refer to in this paper is Laplace LPD which to the best of our knowledge is not limited to gradients of a model and is more broadly applicable as described in [2] for an application to gender privacy.
>
> > Equation (4) gives a bunch of losses with weights. It seems hard to tune those weights on different datasets since there are too many weights. How did you tune them?
>
> We base our choice of weights on previous works. We have clarified the weighting selection in **line 313**.
>
> ---
> [1] Vector-quantized image modeling with improved VQ-GAN
>
> [2] Differentially Private Adversarial Auto-Encoder to Protect Gender in Voice Biometrics

---

### Official Review · Reviewer_Prj4 · 2024-11-08

**Soundness:** 2
**Presentation:** 3
**Contribution:** 1
**Rating:** 3
**Confidence:** 4

**Summary:**

The paper presents a codec-based approach to addressing privacy concerns in speech data used for training large language models. The author claims that the proposed USC is a computationally efficient encoder-decoder model that could disentangle speech into privacy-preserving semantic and residual acoustic and speaker representations.

**Strengths:**

- The privacy-related topic is interesting and the motivation is good.
- The proposed metrics for assessing privacy-preserving properties in speech representations, particularly focusing on k-anonymity, will benefit privacy protection research.

**Weaknesses:**

**Ignore the most related works**
- I believe this work is primarily a speaker anonymization task; however, no speaker anonymization studies are cited in this paper[1-3].

**Limited Contribution**
 - The contribution of this paper is limited, as the disentanglement in the codec model has already been proposed by SpeechTokenizer [4] and FACodec [5] for speech generation tasks. Furthermore, one of the top systems in the VPC 2024 also employed a disentangled codec model for privacy protection [6], though the authors did not cite it. Therefore, the core approach in this paper is not novel.

**Inefficient experiments**
 - VoicePrivacy Challenge (VPC) has been held in three editions[1-3] and aims to provide baseline, evaluation metrics, training and evaluation datasets for speaker anonymization studies. Huge of studies follow the setting of this challenge to evaluate their work, but this paper ignores the most convincing evaluation method, which makes me can not believe the effective of the proposed approach.
   https://www.voiceprivacychallenge.org/

**some of the authors' statements are not rigorous**

 - In the abstract, the author claims that ''Combining both representations, USC achieves state-of-the-art speech reconstruction'', the current experiments can not support this statement, due to some sota disentangled codec models (FaCodec) are not included in the comparison.
 - In the introduction, the statement that ''USC is the lowest bit-rate neural speech codec'' is incorrect, I think WavTokenizer is the lowest, but the author does not cite it in their paper.

**Minor Concern**
- Please simplify Sec. 3.2, GRL method has been widely used in previous studies.
- Please clarify why Hubert L9 without speaker-identifiable traits.
- How to balance the hyper parameters in Eq. 4
- Please PESQ metric in Table 1

[1] The voiceprivacy 2020 challenge: Results and findings

[2] The VoicePrivacy 2022 Challenge: Progress and perspectives in voice anonymisation

[3] The VoicePrivacy 2024 Challenge Evaluation Plan

[4] SpeechTokenizer: Unified Speech Tokenizer for Speech Large Language Models

[5] NaturalSpeech 3: Zero-Shot Speech Synthesis with Factorized Codec and Diffusion Models

[6] NPU-NTU System for Voice Privacy 2024 Challenge

**Questions:**

Could the author explain why the paper does not cite any paper related to speaker anonymization, or clarify the difference between their work and speaker anonymization?

---

> ### Author Response · Authors · 2024-11-21
>
> Thank you for your review and the suggestion of additional literature. Regarding the main question raised:
> > Could the author explain why the paper does not cite any paper related to speaker anonymization, or clarify the difference between their work and speaker anonymization?
>
> The main focus of the work was to maintain speaker privacy, not specifically to completely anonymize the speech. We base our representation learning approach on the assertion in **line 348** and in [1], which states that retaining speech paralinguistic information while effectively achieving perfect anonymization is a conflicting task. Our approach aims to preserve speaker privacy while retaining all the speech paralinguistic characteristics that make speech modeling richer than traditional natural language modeling based on text followed by a TTS model.
>
> Regarding weaknesses:
> > No speaker anonymization studies are cited in this paper[1-3].
>
> Thank you for the suggestion. We have added the required citation for the VPC as it highly correlates with the metrics we propose in Section 4.2. However, the set of privacy-preserving metrics presented comply with the EU regulatory guidelines specified in **line 356**. The metrics we propose are equally informative to the VPC privacy-utility trade-off:
> * The EER metric is an equivalent evaluation to the proposed linkability metric, based on the cosine similarity between non-identifiable samples. We improve it by presenting the singling-out test, its motivation, and correlation with subjective metrics.
> * We also report WER on privacy-preserved reconstructed speech, and we present an equivalent metric to UAR through the Sentiment CCC metric.
>
> > Disentanglement in the codec model has already been proposed by SpeechTokenizer and FACodec for speech generation tasks. Furthermore, one of the top systems in the VPC 2024 also employed a disentangled codec model for privacy protection.
>
> What separates us from other works is that we learn a unified representation for both content and sentiment, simplifying the multi-codebook representations of FACodec and improving on the sentiment encoding of SpeechTokenizer. Again, thank you for the latest released and missing NPU-NTU work. We have added it in our related work section. However, similar to FACodec, NPU-TPU proposes a separate RVQ layer for each attribute with an external speaker conditioning.
>
> > The author claims that ''Combining both representations, _USC_ achieves state-of-the-art speech reconstruction'', the current experiments can not support this statement, due to some sota disentangled codec models (_FaCodec_) are not included in the comparison.
>
> We have decided to add FaCodec as an extra baseline for both high-fidelity and semantic-reconstruction. When we reefer to combining both, we refer to the high-fidelity evaluation, in which USC beats SpeechTokenizer and FaCodec, generating 24 kHz high-fidelity waveforms while disentangling speech characteristics.
>
>
> > In the introduction, the statement that ''USC is the lowest bit-rate neural speech codec'' is incorrect, I think WavTokenizer is the lowest, but the author does not cite it in their paper.
>
> We didn’t include WavTokenizer for two reasons: 1) WavTokenizer was submitted to ICLR for peer-reviewing and work was incomplete when published on ArXiv by end of August, and 2) WavTokenizer falls behind in generating high-quality speech reconstruction with a single codebook, reporting lower PESQ/STOI metrics than full RVQ-based codecs.
>
> Now, addressing the minor concerns:
>
> > Simplify Sec. 3.2, GRL method has been widely used in previous studies.
>
> We still want to keep details on the technique, since we use the AMS Softmax loss function,and other works use speaker reversal on cosine distance of speaker embeddings [2] or plain speaker classification [3, 4].
>
> > Please clarify why Hubert L9 without speaker-identifiable traits.
>
> We describe the choice of L9 HuBERT in **line 232**, where we cite the WavLM ablation showing that L9 contain rich speaker-agnostic semantic information. This is the common layer used in the official HuBERT clustering [5] and in other works like [2, 6].
>
> > How to balance the hyper parameters in Eq. 4
>
> We kept the weights from original works introducing each component. We have clarified the weighting selection in **line 313**.
>
> > Please PESQ metric in Table 1
>
> We have added the PESQ metric to all our baselines. Thanks for the recommendation.
>
> ---
>
> [1] Privacy versus emotion preservation trade-offs in emotion-preserving speaker anonymization
>
> [2] Investigating self-supervised features for expressive, multilingual voice conversion
>
> [3] NaturalSpeech 3: Zero-Shot Speech Synthesis with Factorized Codec and Diffusion Models
>
> [4] Speech Representation Disentanglement with Adversarial Mutual Information Learning for One-shot VC
>
> [5] https://github.com/facebookresearch/fairseq/blob/main/examples/hubert/README.md
>
> [6] Speech Resynthesis from Discrete Disentangled SSL Representations

---

> > ### Comment · Reviewer_Prj4 · 2024-11-24
> >
> > Thanks for the response. But I still have many concerns:
> > - The goal of speaker anonymization is also to preserve speaker privacy while retaining paralinguistic characteristics. In my opinion, the task author mentioned is similar to speaker anonymization.
> > - The proposed metrics may be equally informative to the VPC privacy-utility trade-off, but using metrics from VPC will be more convincing.
> > - DAC outperforms the proposed in all metrics except WER in Table 1, which can not support the statement that ''USC achieves state-of-the-art speech reconstruction''. One more question is why the BW of DAC is 7.75.
> >
> > Therefore, I will leave my score as it is.

---

### Author Response · Authors · 2024-11-21
**Rebuttal revision 1 (paper and samples)**

Dear reviewers,

We would like to express our sincere gratitude for your reviews of our work. We have addressed specific concerns under each official review. With this official comment, we want to inform you that we have uploaded a new revision incorporating the changes mentioned in each specific review.

Overall, addressing common concerns across reviews, we have added FaCodec as another baseline to our model and evaluated it across all our metrics. We have also included an additional PESQ signal quality metric, replaced MCD with ViSQOL, and clarified several explanations that were not sufficiently clear, thanks to your feedback.

Furthermore, we have updated the sample page attached in the Supplementary Material. Given that this work is speech-based, we believe it is crucial to accompany all our tests and evaluations with audio samples to further contextualize and provide audible examples of what we present in this work. We strongly encourage the reviewers, and anyone reading this submission, to listen to these samples.

Thank you for your time,

Best, The authors.

---

### Note · Authors · 2024-10-02

I have read and agree with the venue's withdrawal policy on behalf of myself and my co-authors.

---

> ### Note · Program_Chairs · 2024-10-02
>
> We approve the reversion of withdrawn submission.

---

### Meta-Review · Area_Chair_1Yix · 2024-12-08

**Metareview:**

The paper presents a codec-based approach to addressing privacy concerns in speech data. The proposed method disentangles speech information into two components: semantic information, which includes both the content and paralinguistic aspects of the speech, and speaker information. By separating these elements, the approach enables speech anonymization. The reviewers raised several concerns, including missing important baselines and evaluation metrics. The authors responded to the rebuttal by adding more baselines and evaluation metrics, but did not fully address the concerns.

**Additional Comments On Reviewer Discussion:**

Below is a summary of the concerns raised by reviewers and the authors’ responses:

+ Limited Novelty: SpeechTokenizer (Reviewer Prj4, Reviewer 6QEh) and FACodec (Reviewer Prj4, Reviewer JxJM) have already proposed disentanglement in the codec model. The authors added FACodec as an additional baseline; however, reviewers still expressed concerns regarding the proposed method's performance. The quantitative results were not convincing enough to demonstrate that the proposed method outperforms existing baselines.

+ Incomplete Voice Conversion Experiment: The single example provided was insufficient to demonstrate the quality of voice conversion (Reviewer JxJM, Reviewer vjTT). In response, the authors have uploaded more samples to the sample page.

+ Lack of Evaluation Metrics: Some reviewers (Reviewer YwDt, Reviewer vjTT, Reviewer JxJM) noted a lack of evaluation metrics. Although the authors included PESQ and VISQOL metrics in their revised version, reviewers highlighted the need for subjective listening tests.

+ Speaker Anonymization Research Context: Some reviewers pointed out that significant research has already been conducted on speaker anonymization (e.g., the VoicePrivacy Challenge has been conducted multiple times). The paper lacks references to and comparisons with previous approaches in this area.

---

### Decision · Program_Chairs · 2025-01-22

Reject